# MVSplat360: Feed-Forward 360 Scene Synthesis from Sparse Views

**Yuedong Chen**[1]    **Chuanxia Zheng**[2]    **Haofei Xu**[3,4]    **Bohan Zhuang**[1]
**Andrea Vedaldi**[2]    **Tat-Jen Cham**[5]    **Jianfei Cai**[1]

[1]Monash University    [2]VGG, University of Oxford    [3]ETH Zurich
[4]University of Tübingen, Tübingen AI Center    [5]Nanyang Technological University

## Abstract

We introduce MVSplat360, a feed-forward approach for 360° novel view synthesis (NVS) of diverse real-world scenes, using only sparse observations. This setting is inherently ill-posed due to minimal overlap among input views and insufficient visual information provided, making it challenging for conventional methods to achieve high-quality results. Our MVSplat360 addresses this by effectively combining geometry-aware 3D reconstruction with temporally consistent video generation. Specifically, it refactors a feed-forward 3D Gaussian Splatting (3DGS) model to render features directly into the latent space of a pre-trained Stable Video Diffusion (SVD) model, where these features then act as pose and visual cues to guide the denoising process and produce photorealistic 3D-consistent views. Our model is end-to-end trainable and supports rendering arbitrary views with as few as 5 sparse input views. To evaluate MVSplat360's performance, we introduce a new benchmark using the challenging DL3DV-10K dataset, where MVSplat360 achieves superior visual quality compared to state-of-the-art methods on wide-sweeping or even 360° NVS tasks. Experiments on the existing benchmark RealEstate10K also confirm the effectiveness of our model. Readers are highly recommended to view the video results at `donydchen.github.io/mvsplat360`.

## 1  Introduction

The rapid advancement in 3D reconstruction and NVS has been facilitated by the emergence of differentiable rendering [29, 31, 41, 40, 21]. These methods, while fundamental and impressive, are primarily tailored for per-scene optimization, requiring hundreds or even thousands of images to comprehensively capture every aspect of the scene. Consequently, the optimization process for each scene can be time-consuming, and collecting thousands of images is impractical for casual users.

In contrast, we consider the problem of novel view synthesis in diverse real-world scenes using a limited number of source views through a feed-forward network. In particular, this work investigates *the feasibility of rendering wide-sweeping or even 360° novel views using extremely sparse observations*, like fewer than 5 images. This task is inherently challenging due to the complexity of scenes, where the limited views do not contain sufficient information to recover the whole 3D scene. Consequently, there is a necessity to ensemble visible information under minimal overlap accurately and generate missing details reasonably.

This represents a new problem setting in sparse-view feed-forward NVS. Existing feed-forward methods typically focus on two distinct scenarios: 360° NVS with extremely sparse observations, but only at *object-level* [20, 52, 66, 59, 27, 72, 63, 56, 62, 50, 46, 47, 18], or generating reasonable results for *scene-level* synthesis, but only for nearby viewpoints [53, 7, 19, 43, 9, 13, 6, 60, 10, 70, 42, 61]. In contrast, we argue that the time is ripe to unify these previously distinct research directions.

38th Conference on Neural Information Processing Systems (NeurIPS 2024).

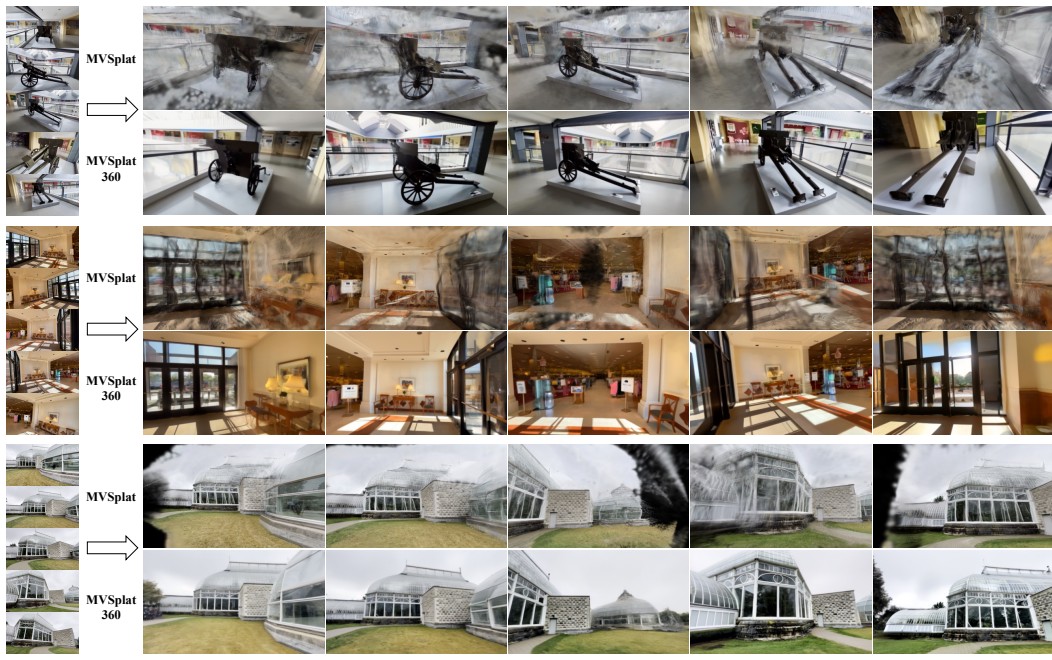

Figure 1: **Examples of our MVSplat360**. Given sparse and wide-baseline observations of diverse in-the-wild scenes, MVSplat360 can directly render 360° novel views (inward or outward facing) or other natural camera trajectory views in a *feed-forward* manner, without any per-scene optimization.

Our goal should be to develop systems capable of synthesizing wide-sweeping or even 360° novel views of large, real-world scenes with complex geometry and significant occlusion. Specifically, this work explores synthesising 360° novel views from fewer than 5 input images. We show that in this challenging setting, existing feed-forward scene synthesis approaches [9, 13, 6, 60, 10, 57] struggle to succeed. This failure arises from two main factors: i) the limited overlap among input views causes many contents to appear in only a few views or even a single one, posing significant challenges for 3D reconstruction; ii) the extremely sparse observations lack sufficient information to capture the comprehensive details of the whole scene, resulting in regions unobserved from novel viewpoints.

In this paper, we propose a simple yet effective framework to address these limitations and introduce the first benchmark for feed-forward 360° scene synthesis from sparse input views. Our key idea is to leverage prior knowledge from a large-scale pre-trained latent diffusion model (LDM) [35] to "imagine" plausible unobserved and disoccluded regions in novel views, which are inherently highly ambiguous. Unlike existing 360° object-level NVS approaches [27, 63, 49, 56, 25, 72, 50], large-scale real-world scenes comprise multiple 3D assets with *complex arrangements, heavy occlusions*, and *varying rendering trajectories*, which makes it particularly challenging to condition solely on camera poses, as also verified by concurrent work ViewCrafter [68].

To develop a performant framework for scene-level synthesis, we opt to treat the LDM as a refinement module, while relying on a 3D reconstruction model to process the complex geometric information. Broadly, we build upon the feed-forward 3DGS [21] model, MVSplat [10], to obtain coarse novel views by matching and fusing multi-view information with the cross-view transformer and cost volume. Although these results are imperfect, exhibiting visual artifacts and missing regions (see Fig. 1), they represent the reasonable geometric structure of the scene, as they are rendered from 3D representation. Furthermore, we choose Stable Video Diffusion (SVD) [3] over other image-based LDM as the refinement module, since its strong temporal consistency capabilities align better with the view-consistent requirement of the NVS task, as also observed by concurrent work 3DGS-Enhancer [28]. Conditioning SVD with the 3DGS rendered outputs, our MVSplat360 produces visually appealing novel views that are multi-view consistent and geometrically accurate (see Fig. 1).

Importantly, the original MVSplat outputs only RGB images, which is not the optimal condition for the generator, and is difficult to optimize jointly with the SVD denoising module. To tackle this, we

propose a simple Gaussian feature rendering with multi-channels, supervised with an introduced latent space alignment loss. Despite a seemingly minor change, the additional feature condition for SVD leads to a significant impact: It bypasses the SVD's frozen image encoder, allowing the gradients from SVD to backpropagate to enhance the geometry backbone and lead to improved visual quality, especially on the new challenging DL3DV-10K dataset. While related work Reconfusion [58], CAT3D [15] and latentSplat [57] also combine the 3D representation with 2D generators, the former two focus more on per-scene optimisation, while the latter only shows 360° NVS at the object level.

We conduct a series of experiments, mainly on two datasets. First, we establish a new benchmark on DL3DV-10K dataset [23], creating a new training and testing split for feed-forward wide-sweeping and 360° NVS. In this challenging setting, our MVSplat360 achieves photorealistic 360° NVS from sparse observations and demonstrates significantly better visual quality, where the previous scene-level feed-forward methods [9, 6, 60, 10] fail to achieve plausible results. Second, we deploy MVSplat360 on the existing RealEstate10K [74] benchmark. Following latentSplat [57], we estimate both interpolation and extrapolation NVS, and report state-of-the-art performance.

Our main contributions can be summarized as follows. 1) We introduce a crucial and pressing problem for novel view synthesis, *i.e.*, how to do wide-sweeping or even 360° NVS from sparse and widely-displaced observations of diverse in-the-wild scenes (*not objects*) in a feed-forward manner (*no any per-scene optimization*). 2) We propose an effective solution that nicely integrates the latest feed-forward 3DGS and the pre-trained Stable Video Diffusion (SVD) model with meticulous integration designs, where the former is for reconstructing coarse geometry and the latter is for refining the noisy and incomplete coarse reconstruction. 3) Extensive results on the challenging DL3DV-10K and RealEstate10K datasets demonstrate the superior performance of our MVSplat360.

## 2 Related Work

**Sparse view per-scene reconstruction and synthesis.** Differentiable rendering methods, such as NeRF [31] and 3DGS [21], are mainly designed for very dense views (*e.g.*, 100) as inputs for per-scene optimization, which is impractical to collect for casual users in real applications. To bypass the requirement for dense views, various regularization terms have been proposed in per-scene optimization [33, 11, 69, 48]. Recently, ZeroNVS [38], Reconfusion [58] and concurrent submissions, including CAT3D [15], ReconX [24], ViewCrafter [68], LM-Gaussian [67], 3DGS-Enhancer [28], have leveraged large-scale diffusion models for generating pseudo dense views of a 3D scene, which are then input into a per-scene reconstruction pipeline. However, these methods are inherently slow for reconstructing unseen scenes due to the necessity of per-scene optimisation.

**Feed-forward scene reconstruction and synthesis.** To mitigate these limitations, early approaches like Light Field Networks [40] use ray querying to predict novel views. Subsequent methods [44, 43] employ epipolar attention for multi-view geometry estimation. Later, pixelNeRF [66] devised pixel-aligned features for NeRF reconstruction [31], leading to a range of subsequent methods that incorporate feature matching fusion [7, 9], Transformers [36, 13, 32] and 3D volume representation [7, 60]. Recently, 3D Gaussian Splatting [21] has been implemented into feed-forward networks, such as pixelSplat [6], MVSplat [10], Splatter Image [46], Flash3D [45] and latentSplat [57]. While these methods were successful in novel view synthesis from sparse views, they fail to achieve this in a wide-sweeping or 360° setting. Concurrent yet unpublished submissions, DepthSplat [61] and Long-LRM [75], also show promising results in the 360° setting, but their frameworks have limited generation capabilities, necessitating the use of denser inputs, *e.g.*, 12 or 32 views.

**Camera trajectory controllable synthesis.** Generative models have achieved remarkable results for image/video synthesis [14, 73, 8, 35, 5, 17, 39, 3], but they lack precise control over the viewpoint of generated images. To address this, several approaches fine-tune large-scale pre-trained diffusion models with explicit image and pose conditions [27, 26, 25, 4, 72, 64, 56, 50]. However, these methods mainly show 360° NVS results on single objects, leaving the complex scene synthesis problem unsolved. Natural scenes comprise multiple objects with intricate occlusion relationships, presenting greater challenges that are not easily addressed by these single-object NVS models. Besides, camera trajectories can be highly irregular and varied when roving around such complex scenes. Although related works [55, 22, 38, 15] have explored training or fine-tuning diffusion models with camera control for scene synthesis, they often struggle with precise camera pose control [55, 22], and still rely on per-scene optimization for 3D reconstruction [38, 15].

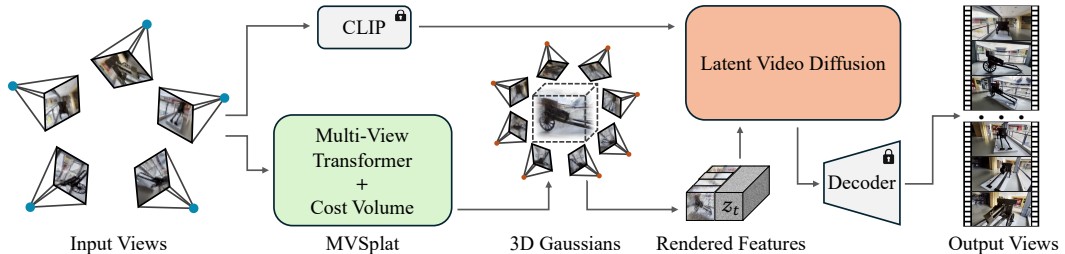

| Input Views | MVSplat | 3D Gaussians | Rendered Features | Output Views |

Figure 2: **Overview of our MVSplat360**. (a) Given sparse posed images as input, we first match and fuse the multi-view information using a multi-view Transformer and cost volume-based encoder. (b) Next, a 3DGS representation is constructed to represent the coarse geometry of the entire scene. (c) Considering such coarse reconstruction is imperfect, we further adapt a pre-trained SVD, using features rendered from the 3DGS representation as conditions to achieve 360° novel view synthesis.

## 3 Methodology

Given $N$ sparse views $\mathcal{I} = \{\boldsymbol{I}^i\}_{i=1}^N$ and the corresponding camera poses $\mathcal{P} = \{\boldsymbol{P}^i\}_{i=1}^N$, with $\boldsymbol{P}^i = (\mathbf{K}^i, \mathbf{R}^i, \mathbf{T}^i)$ comprising intrinsic $\mathbf{K}^i$, rotation $\mathbf{R}^i$ and translation $\mathbf{T}^i$, our goal is to learn a model $\Phi$ that synthesizes wide-sweeping or even 360° novel view synthesis (NVS).

We opt to go beyond per-scene optimisation [1, 2, 58, 15], and to deal with a more general feed-forward network capable of achieving 360° NVS for unseen scenes, *yet without the need of additional per-scene training*. This requires effectively matching information between sparse views in 3D space, as well as generating sufficient content based on only partial observations. To achieve that, our MVSplat360 framework, illustrated in Fig. 2, comprises two main components: a multi-view geometry reconstruction module (Section 3.1) and a multi-frame consistent appearance refinement network (Section 3.2). The former is responsible for matching and fusing multi-view information from sparse observations to create a coarse geometry reconstruction, whereas the latter is designed to refine the appearance with a pre-trained latent video diffusion model. While similar two-step approaches have been explored in recent related works, *e.g.*, [4, 58, 38, 15], we are the first (to the best of our knowledge) to explore it on wide-sweeping or even 360° NVS for large-scale scenes from sparse views (as few as 5), *in a feed-forward manner*.

### 3.1 Multi-View Coarse Geometry Reconstruction

The first module is built upon a feed-forward 3DGS reconstruction model, *i.e.*, MVSplat [10] as in our implementation. Specifically, given sparse-view observations $\mathcal{I} = \{\boldsymbol{I}^i\}_{i=1}^N$ and their corresponding camera poses $\mathcal{P} = \{\boldsymbol{P}^i\}_{i=1}^N$, the model learns to predict 3D Gaussian parameters $\{(\boldsymbol{\mu}_i, \alpha_i, \boldsymbol{\Sigma}_i, \boldsymbol{c}_i)\}_{i=1}^{H \times W \times N}$, which can then be splatted to obtain a set of RGB images $\tilde{\mathcal{I}}^{\text{tgt}}$ using the target camera poses $\mathcal{P}^{\text{tgt}}$. To ensure better integration with the following diffusion module, we predict an additional Gaussian feature $\hat{\boldsymbol{f}}_i$, in parallel with other parameters, which can be rasterized to the corresponding latent features $\hat{\mathcal{F}}^{\text{tgt}}$. Furthermore, we also improve the view selection strategy to improve the model's robustness in handling widely displaced inputs.

**Coarse geometry reconstruction.** Our backbone comprises multi-view feature extraction, cost volume construction, depth estimation, and 3D Gaussian parameter predictions. First, a cross-view transformer encoder is applied to fuse multi-view information and obtain cross-view aware features $\mathcal{F} = \{\boldsymbol{F}^i\}_{i=1}^N$. Then, $N$ cost volumes $\mathcal{C} = \{\boldsymbol{C}^i\}_{i=1}^N$ are constructed by matching feature correlations between cross-views. Specifically, it uniformly divides the depth into $L$ layers in the near and far depth ranges, *i.e.* $\mathcal{D} = \{D_m\}_{m=1}^L$, and then warps the features from one view $j$ to another view $i$ via $\boldsymbol{F}_{D_m}^{j \to i} = \mathcal{W}(\boldsymbol{F}^j, \boldsymbol{P}^i, \boldsymbol{P}^j, D_m)$. The cost volume $\boldsymbol{C}^i = [\boldsymbol{C}_{D_1}^i, \boldsymbol{C}_{D_2}^i, \dots, \boldsymbol{C}_{D_L}^i]$ is then collected by $L$ correlations, where each correlation is expressed as $\boldsymbol{C}_{D_m}^i = \frac{\boldsymbol{F}_{D_m}^{j \to i} \cdot \boldsymbol{F}^i}{\sqrt{C}}$, with $C$ denoting channel dimension. Finally, the per-view estimated depth $d$ is obtained by applying the `softmax` operation on the cost volumes in the depth dimension. After that, the Gaussian mean is computed by $\boldsymbol{\mu} = \mathbf{K}^{-1}\boldsymbol{u}d + \Delta$, where $\mathbf{K}$ is the camera intrinsic, $\boldsymbol{u} = (u_x, u_y, 1)$ denotes each pixel, and $\Delta \in \mathbb{R}^3$ is the predicted offset, along with opacity $a \in [0, 1]$, covariance $\Sigma \in \mathbb{R}^{3 \times 3}$, and color $\boldsymbol{c} \in \mathbb{R}^{3(S+1)^2}$,

where $S$ is the order of the spherical harmonics [10]. Once the model predicts a set of 3D Gaussian parameters $\{(\boldsymbol{\mu}_i, \alpha_i, \boldsymbol{\Sigma}_i, \boldsymbol{c}_i)\}_{i=1}^{H \times W \times N}$, the target view $\tilde{\mathcal{I}}^{\text{tgt}}$ can be rendered through rasterization.

**Gaussian feature rendering.** Given sparse-view observations, MVSplat [10] tends to render images with noticeable artifacts in wide-sweeping novel viewpoints (see Fig. 1), resulting in suboptimal conditioning for the subsequent SVD. Since the rendered images must first be encoded into the latent space using a frozen encoder (see Section 3.2), enhancing the backbone with gradients from SVD would be computationally expensive. To address this issue, we propose directly rasterising features $\hat{\mathcal{F}}$ into *the latent space* of SVD, by predicting an additional parameter $\hat{\boldsymbol{f}}_i$ for each 3D Gaussian. This operation offers two advantages: (i) The latent feature includes multi-channel information, providing a more comprehensive representation of the scene; (ii) The entire framework is end-to-end connected by conditioning SVD on the rendered latent features instead of the image-encoded ones. It enables the SVD loss to optimize the Gaussian features, further enhancing the reconstruction backbone.

**Observed and novel viewpoints selection.** To enable 360° scene synthesis, it is crucial to choose the correct camera viewpoints, so that they can cover most contents in diverse and complex scenes [15]. It is impractical to assume a circular orbital camera trajectory like those object-level 360° view synthesis [27, 46, 72, 47], whereas it is suboptimal to randomly choose a video sequence like existing scene-level nearby viewpoint synthesis [46, 6, 10]. To this end, we propose to choose views *evenly distributed* within a set of targeted viewpoints as input. Specifically, for a given set of candidate views, we apply farthest point sampling over the camera locations to identify the input views and randomly choose from the rest as target views. The number of candidate views gradually increases throughout the training, stably improving the model's capability toward handling 360° scene synthesis.

**View interaction within the local group.** Recalling that the 3D reconstruction backbone MVSplat is primarily designed for nearby viewpoints, with key components like multi-view transformers and cost volume assuming sufficient overlap among input views. However, in the more challenging 360° settings, the widely displaced input views lead to minimal overlap between specific view pairs, hindering the effectiveness of the backbone. To mitigate this limitation, we refactor our backbone to use cross-view attention and construct the cost volume *only within a local group* of input views based on camera locations, reducing memory consumption and ensuring stable model convergence.

## 3.2 Multi-Frame Appearance Refinement

**Video diffusion model.** MVSplat360 utilizes an off-the-shelf multi-frame diffusion model, *i.e.* Stable Video Diffusion (SVD) [3], to refine the visual appearance of the aforementioned coarse reconstruction. SVD is pre-trained on large-scale video datasets and has strong prior knowledge of temporal consistency. It adheres to the original formulation used by Stable Diffusion (SD) [35] that conducts the denoising processing in the latent space. In particular, given a target sequence of $x^{1:M}$ with $M$ images, they are initially embedded into the latent space by a frozen encoder $\mathcal{E}$, yielding $z_0^{1:M} = \mathcal{E}(x^{1:M})$, and then perturbed by adding Gaussian noise $\epsilon \sim \mathcal{N}(0, \boldsymbol{I})$ in a Markov process:

$$z_t^{1:M} = \sqrt{\bar{\alpha}_t} z_0^{1:M} + \sqrt{1 - \bar{\alpha}_t} \epsilon_t^{1:M}, \tag{1}$$

while $\bar{\alpha}_0, \ldots, \bar{\alpha}_T$ is a pre-defined noise schedule within $T$ steps. Given noise input $z_t^{1:M}$, the denoiser $\epsilon_\theta$ is then trained by optimizing the following objective function:

$$\min_{\epsilon_\theta} \mathbb{E}_{(x^{1:M}, y), t, \epsilon^{1:M} \sim \mathcal{N}(0,1)} \left[ \| \epsilon^{1:M} - \epsilon_\theta(z_t^{1:M}, t, y) \|_2^2 \right], \tag{2}$$

where $x^{1:M}$ is the target images, and $y$ is the conditional inputs. After $\epsilon_\theta$ is trained, the model can generate a video by performing iterative denoising from pure Gaussians $z_T^{1:M}$ conditioned on $y$.

Note that SVD is trained with the latest **v**-prediction formulation [37], instead of the original $\epsilon$-prediction in SD. Hence, the final loss is calculated in latent space using the mean squared error (MSE) between the ground truth and its prediction $\|z_0^{1:M} - \hat{z}_t^{1:M}\|_2^2$, where $\hat{z}_t^{1:M}$ is obtained by translating the velocity $\mathbf{v} = \Phi(z_t^{1:M}, t, y)$ to latent space, *i.e.*, $\hat{z}_t^{1:M} = \alpha_t z_t^{1:M} - \sigma_t \mathbf{v}$.

**Multi-view hybrid conditions.** To ensure an accurate understanding of the scene, the model requires the integration of both low-level perception (*e.g.*, depth and texture) and high-level understanding (*e.g.*, semantics and geometry). Following [27, 72, 50], we adopt a hybrid conditioning mechanism to fine-tune the SVD model for wide-sweeping NVS with sparse observations.

In one stream, a CLIP [34] image embedding token of the original visible views $\mathcal{I}$ is used as a global type and text prompt. At each UNet block, a cross-attention operation is applied to capture high-level

semantics of the input images to the model. Since we have sparse views, we average these tokens to become one global token. In the other stream, the spatial conditions from the coarse geometry rendered features $\hat{\mathcal{F}}^{\text{tgt}} = \{\hat{F}_i\}_{i=1}^{M}$ is channel-concatenated with the noised latent $z_t^{1:M}$. These spatially conditional features assist the model to capture the view information, and learn low-level perception to maintain the texture of the scenes. Compared to the concurrent work CAT3D [15], this coarse feature conditioning not only provides accurate pose information from the 3DGS rendering, but also offers reasonable visual information.

**Color adjustment.** While our MVSplat360 can achieve photorealistic NVS, the synthesized videos sometimes exhibit oversaturated colors (detailed in Appendix C). This may be visually acceptable for video generation, but it can decrease performance when evaluated on NVS task. To mitigate this, we apply post-processing by matching the color histogram between the SVD refined views $\hat{\mathcal{I}}^{\text{tgt}}$ and our 3DGS rendered views $\tilde{\mathcal{I}}^{\text{tgt}}$ before performing pixel-aligned measurements, *i.e.*, PSNR and SSIM.

### 3.3 Training Objectives

Our MVSplat360 predicts two sets of images, including the coarse one $\tilde{\mathcal{I}}^{\text{tgt}}$ from the 3DGS module and the refined one $\hat{\mathcal{I}}^{\text{tgt}}$ from the SVD module, where the former is mainly rendered to help supervise the geometry backbone. The entire model is end-to-end trainable, using three groups of loss functions, namely reconstruction loss, diffusion loss and latent space alignment loss.

In particular, the reconstruction loss is a linear combination of $\ell_2$ and LPIPS [71], applied between the coarse outputs $\tilde{\mathcal{I}}^{\text{tgt}}$ and the corresponding ground truth $\mathcal{I}^{\text{tgt}}$. The other two loss functions are applied to the following SVD module, whose gradients will backpropagate to the 3DGS module but will not update those structural parameters, *i.e.*, $\boldsymbol{\mu}, \alpha, \boldsymbol{\Sigma}$. This is achieved by stopping the gradients from the structural parameters when rendering the latent features $\tilde{\mathcal{F}}^{\text{tgt}}$, since keeping those gradient flows will lead to unstable training, as also observed by latentSplat [57]. We use the standard v-prediction formulation (detailed in Section 3.2) as the diffusion loss to fine-tune the denoising network of the SVD, keeping the first-stage encoder and decoder frozen. Since the SVD's released model is conditioned on image-encoded features while our diffusion module is on 3DGS-rendered ones, we find it beneficial to align these two spaces by regularizing with a latent space alignment loss, $\min_{g_\theta} \mathbb{E}_{\hat{\mathcal{F}} \sim g(\mathcal{I})} \|\mathcal{E}(\mathcal{I}^{\text{tgt}}) - \hat{\mathcal{F}}^{\text{tgt}}\|_2^2$, where $g$ refers to the geometry backbone with trainable parameters $\theta$ and $\mathcal{E}$ is the frozen SVD encoder.

## 4 Experiments

### 4.1 Experimental Details

**Datasets.** To verify the effectiveness of MVSplat360 in synthesizing wide-sweeping and 360° novel views, we have established a challenging benchmark derived from DL3DV-10K [23]. It comprises 51.3 million frames from 10,510 real-world scenes, adhering to 65 point-of-interest (POI) [65] categories. For training, we use a subset in subfolders "3K" and "4K", resulting in ~2,000 scenes. We tested on the 140 benchmark scenes and filtered them out from the training set to ensure correctness. For each scene, we selected 5 input views using farthest point sampling based on camera locations and evaluated 56 views by equally sampling from the remaining, yielding a total of 7,840 test views. Additionally, since most DL3DV-10K scenes contain a two-round trajectory, we also report another setting by focusing only on half of the sequence, intending to cover the camera trajectory of one round. We denote the two settings as $n = 300$ and $n = 150$, where $n$ refers to the frame distance span across all test views, as most scenes contain roughly 300 frames. We also assess our model on RealEstate10K [74], which contains real estate videos downloaded from YouTube. Consistent with existing works [6, 10], we train MVSplat360 on 67,477 scenes and test it on 7,289 scenes.

**Metrics.** To measure models from different perspectives, we follow [57] to report both the pixel-align metrics, *i.e.*, PSNR and SSIM [54], and the perceptual metrics, *i.e.*, LPIPS [71] and DISTS [12]. Since MVSplat360 aims to generate plausible contents for unobserved and disoccluded regions, we also reported the distribution metric, *i.e.*, Fréchet Inception Distance (FID) [16][1], which measures the similarity between distributions of the generated images and the real ones.

---

[1] `https://github.com/mseitzer/pytorch-fid`

Table 1: **Comparison with SoTA methods on DL3DV-10K**. Below, $n$ is the frame distance span across all the tested novel views within each scene, which is set to 300 by default as most DL3DV-10K scenes contain roughly 300 extracted frames. Since most DL3DV-10K scenes contain a two-round trajectory, we also report another setting of $n = 150$ aiming for coverage of one round.

| Method | $n = 300$ | | | | | $n = 150$ | | | | |
|---|---|---|---|---|---|---|---|---|---|---|
| | PSNR↑ | SSIM↑ | LPIPS↓ | DISTS↓ | FID↓ | PSNR↑ | SSIM↑ | LPIPS↓ | DISTS↓ | FID↓ |
| pixelSplat [6] | 14.83 | 0.401 | 0.576 | 0.383 | 142.83 | 16.05 | 0.453 | 0.521 | 0.348 | 134.70 |
| MVSplat [10] | 15.72 | 0.433 | 0.501 | 0.291 | 78.95 | 17.05 | 0.499 | 0.435 | 0.247 | 61.92 |
| latentSplat [57] | 16.68 | 0.469 | 0.439 | 0.234 | 37.68 | 17.79 | 0.527 | 0.391 | 0.206 | 34.55 |
| MVSplat360 | **16.81** | **0.514** | **0.418** | **0.175** | **17.01** | **17.81** | **0.562** | **0.352** | **0.151** | **18.89** |

**Implementation details.** MVSplat360 is implemented with PyTorch and a CUDA-implemented 3DGS renderer. For coarse geometry reconstruction (Section 3.1), we set hyperparameters following MVSplat [10], except that we apply cross-view attention and build each cost volume within the nearest 2 views rather than all other views. For multi-frame appearance refinement (Section 3.2), we fine-tune from the 14-frame SVD [3] pre-trained model, but using rendered Gaussian features as conditions. We also remove the original "motion value" and "fps" conditions, since they are unrelated to our NVS task. We rescale the rendered feature to have a similar shape as the original image-encoded latent feature in the pre-trained model, which is critical for getting better details as it affects the decoder (more discussions are in Appendix A). We train SVD using 14 frames sampled along natural camera trajectories, captured by the initial videos. At inference, we directly feed 56 views to SVD but change all related temporal attention blocks to local attention with a window size of 14 to better align with the training. More implementation details can be found in Appendix B, and the codes are publicly available at `https://github.com/donydchen/mvsplat360`.

## 4.2 Results on the New DL3DV-10K Benchmark

We first assess the ability of MVSplat360 and baselines to synthesize wide-sweeping and 360° NVS in the newly constructed challenging benchmark with diverse scene categories.

**Baselines.** We perform a thorough comparison of MVSplat360 to the latest state-of-the-art (SoTA) 3DGS-based models, including pixelSplat [6], MVSplat [10] and latentSplat [57]. All models are trained on the same training split and evaluated on the publicly available 140 scenes.

**Quantitative results.** All models are trained to 100K steps and reported at Table 1, except for the latentSplat, which suffers from unstable training due to its GAN-based architecture. Hence, we report its best performance at around 60K training steps before the subsequent collapse. Our MVSplat360 outperforms all existing SoTA models in all metrics on the two settings $n = 300$ and $n = 150$. All models generally perform better on the $n = 150$ setting than on the $n = 300$ one since the latter spans larger viewpoints. It can be seen that the two generative models (latentSpalt and MVSplat360) generally perform better than the other two regression models, suggesting the importance of additional refinement in addressing feed-forward scene reconstruction.

Although our improvement on pixel-aligned metrics appears minor, this is expected since refinement via either interpolation (for disoccluded regions) or extrapolation (for unobserved regions) does not guarantee matching the ground truth at the pixel level. It mainly aims to provide a reasonable solution to refine the images and ensure they align with real-world image distribution. This is verified by the fact that our improvements on perceptual metrics are larger, and it is even more apparent on FID, which measures the distribution deviation. The superiority of our MVSplat360 stands out more from the qualitative results presented below.

**Qualitative results.** The qualitative comparisons are visualized in Fig. 3. MVSplat360 achieves remarkable visual results even under challenging conditions. pixelSplat [6] and MVSplat [10] exhibit obvious artifacts due to the issue of floating Gaussians. latentSplat [57] improves the results with an additional decoder and adversarial training. However, its resulting object geometry and image quality are still far from satisfactory, suggesting that the GAN-based framework cannot provide enough prior knowledge for refining 360° NVS in diverse real-world scenes. Readers are referred to our project page for video results with more comprehensive comparisons, where our MVSplat360 shows

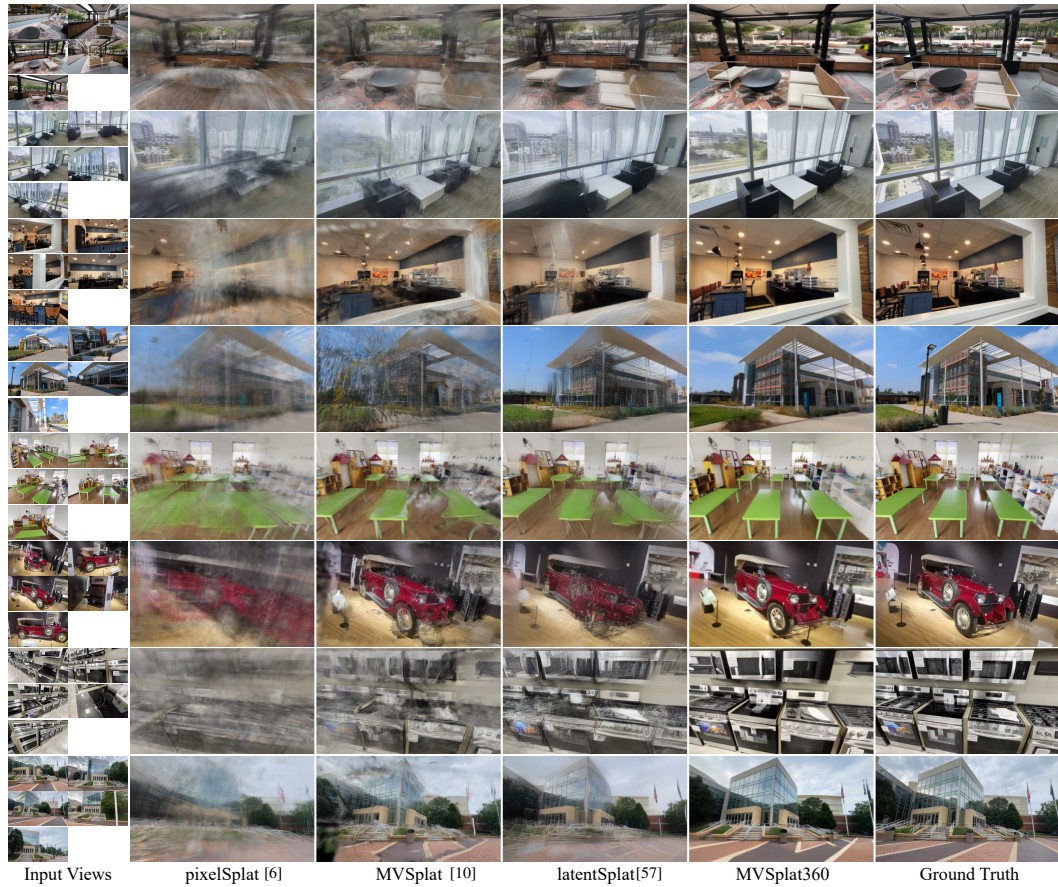

| Input Views | pixelSplat [6] | MVSplat [10] | latentSplat[57] | MVSplat360 | Ground Truth |

Figure 3: **Qualitative comparisons on DL3DV-10K**. MVSplat360 shows significant improvement compared to existing SoTA models. Here, we showcase with a rich mix of diversity and complexity, including indoor (bounded) *vs.* outdoor (unbounded), high *vs.* low texture frequency, more *vs.* less reflection, and more *vs.* less transparency. More results are provided in Appendix E.

Table 2: **Comparison with SoTA methods on RealEstate10K**. We report interpolation scores using the settings of [6, 10], where we retrain latentSplat [57] to maintain fair comparison (indicates with [*]). We report extrapolation scores by following [57].

| Method | Interpolation | | | Extrapolation | | | | |
|---|---|---|---|---|---|---|---|---|
| | PSNR↑ | SSIM↑ | LPIPS↓ | PSNR↑ | SSIM↑ | LPIPS↓ | DISTS↓ | FID↓ |
| PixelNeRF [66] | 20.43 | 0.589 | 0.550 | 20.05 | 0.575 | 0.567 | 0.371 | 160.77 |
| Du *et al.* [13] | 24.78 | 0.820 | 0.213 | 21.83 | 0.790 | 0.242 | 0.144 | 11.34 |
| pixelSplat [6] | 25.89 | 0.858 | 0.142 | 21.84 | 0.777 | 0.216 | 0.130 | 5.78 |
| latentSplat[*] [57] | 25.53 | 0.851 | 0.139 | 22.62 | 0.777 | 0.196 | 0.109 | 2.79 |
| MVSplat [10] | 26.39 | **0.869** | 0.128 | 23.04 | **0.812** | 0.185 | 0.110 | 3.83 |
| MVSplat360 | **26.41** | 0.869 | **0.126** | **23.16** | 0.810 | **0.176** | **0.104** | **1.79** |

multi-view consistent, high-quality visual results along complex camera trajectories, while others suffer from apparent artifacts.

### 4.3 Results on the Existing RealEstate10K Benchmark

We also assess MVSplat360 on the existing benchmark, following the existing "Interpolation" setting [6, 10] and "Extrapolation" setting [57]. We retrained latentSplat on Interpolation and MVSplat on Extrapolation for fair comparisons.

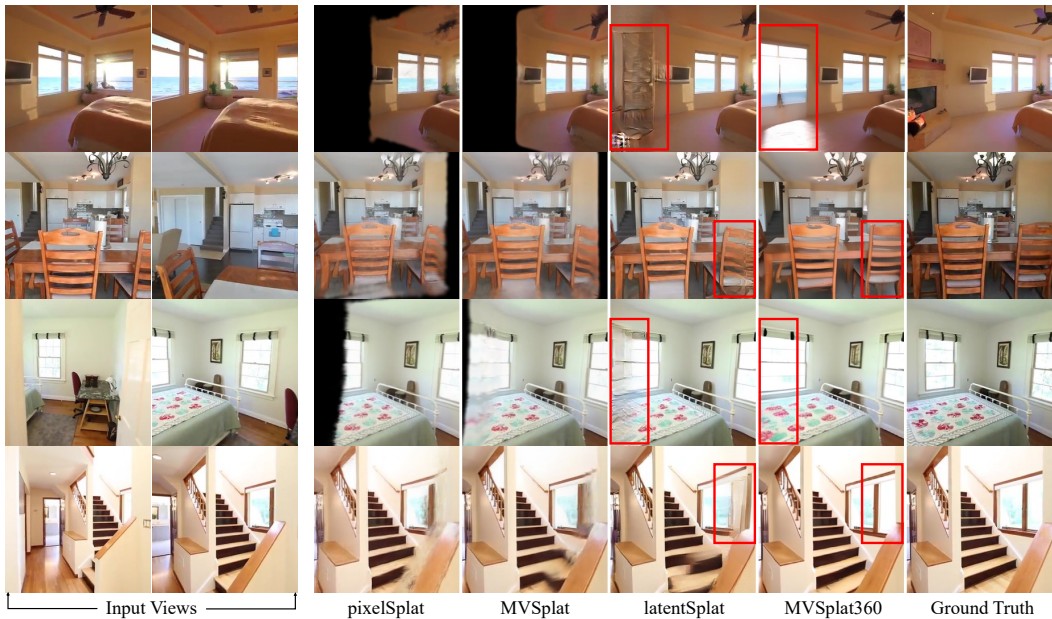

| Input Views | pixelSplat | MVSplat | latentSplat | MVSplat360 | Ground Truth |

Figure 4: **Qualitative comparisons on RealEstate10K**. MVSplat360 shows reasonable generations for disoccluded and unobserved regions, while latentSplat [57] fills in content with artifacts.

Table 3: **MVSplat360 ablations**. All models are trained and evaluated on the DL3DV-10K dataset.

| Models | SSIM↑ | LPIPS↓ | DISTS↓ | FID↓ |
|---|---|---|---|---|
| Baseline | 0.433 | 0.501 | 0.291 | 78.95 |
| + SVD | 0.399 | 0.556 | 0.248 | 38.05 |
| + ctx-attn | 0.467 | 0.451 | 0.185 | 22.78 |
| + GS-feat. | **0.514** | **0.418** | **0.175** | **17.01** |

(a) **Model components**. The baseline is the original MVSplat [10]. 'ctx-attn' refers to using multiple context views to enhance the cross attention, while 'GS-feat.' is the default model where SVD is conditioned with the 3DGS rendered features.

| Views | SSIM↑ | LPIPS↓ | DISTS↓ | FID↓ |
|---|---|---|---|---|
| 3 views | 0.432 | 0.485 | 0.203 | 21.40 |
| 4 views | 0.464 | 0.448 | 0.187 | 18.79 |
| default | 0.514 | 0.418 | 0.175 | 17.01 |
| 6 views | 0.514 | 0.401 | 0.169 | 16.54 |
| 7 views | **0.526** | **0.390** | **0.166** | **16.26** |

(b) **Number of input views**. The 'default' model is trained and tested with 5 views, while the others are directly evaluated with different numbers of input views during testing.

**Quantitative results.** Table 2 shows quantitative comparisons on RealEstate10K [74] of MVSplat360 and other approaches. Our MVSplat360 surpasses all previous state-of-the-art methods, mainly in terms of the perceptual metrics and the distribution metric. The former implies that our rendered views are more aligned with human perception, while the latter shows that our refined images correspond better to the dataset distribution. These observations can be further confirmed by visual assessment.

**Qualitative results.** The qualitative comparisons of the top four best models are in Fig. 4. Pixel-Splat [6] and MVSplat [10] fail to render any content for the unobserved regions due to the lack of generative capability. In contrast, latentSplat can perform extrapolation via its GAN-based decoder, improving the overall visual quality. However, we observed that the content generated by latentSplat is not visually reasonable. Our MVSplat360 generates more plausible content (see the "window" in 1st row and "chair" in 2nd row), thanks to the stronger generative capability of the diffusion model.

### 4.4 Ablations and Analysis

We conduct ablations on DL3DV-10K [23] to analyze MVSplat360 further. Results are reported in Table 3, and discussed in detail next.

**Accessing model components.** The baseline refers to MVSplat [10] since our model is built on top of it. (i) A natural extension is to render novel views from MVSplat and use them directly as conditions

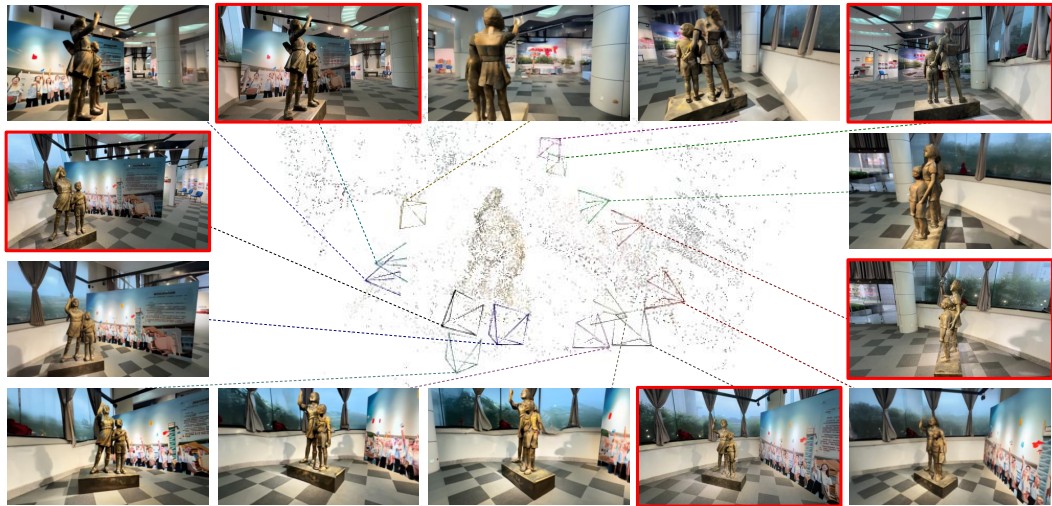

Figure 5: **SfM on input and rendered views**. Images with red borders are the input views, while others are rendered by our MVSplat360. The reasonably recovered camera poses and 3D point clouds via VGGSfM imply that our outputs are multi-view consistent and geometrically correct.

in the SVD denoising process. However, this straightforward approach performs slightly worse than the original MVSplat, likely because SVD struggles to infer pose and visual cues from the noisy image-encoded features. (ii) To better utilize the input context views, we average CLIP-embedded tokens from all views instead of just the first. This provides SVD with richer scene information via the cross-attention blocks, leading to a noticeable improvement. (iii) Lastly, we render high-dimensional features via the 3DGS rasterizer and concatenate them directly into the diffusion latent space, enabling gradients from the denoising UNet to backpropagate through the geometry backbone. This end-to-end training improves performance significantly and is used in our default model.

**Accessing the number of input views.** As shown in Table 4b, while our model is only trained with 5 input views, the performance can be gradually improved by adding more input views at testing. This is reasonable as more input views can provide more observable areas. On the contrary, reducing input views will inevitably result in worse performance. Surprisingly, even with 3 sparse views, our MVSplat360 still outperforms regression models (pixelSplat and MVSplat) that use 5 views.

**Assessing the geometry accuracy.** Our MVSplat360 builds on the video diffusion model SVD, which does ensure strong temporal/multi-frame consistency but does not inherently guarantee geometric accuracy. To confirm that our MVSplat360 produces geometrically accurate outputs, we run structure-from-motion (SfM) on both the input source views and the rendered novel views using VGGSfM [51]. As shown in Fig. 5, VGGSfM recovers reasonable camera poses and 3D point clouds, confirming that our novel views are both multi-view consistent and geometrically correct. This highlights how the 3DGS backbone's latent features provide essential geometric cues, enhancing 3D consistency in the final SVD-based outputs.

## 5 Conclusion

We present MVSplat360, a feed-forward model that synthesises 360° novel views of diverse real-world scenes from sparse input views. Our MVSplat360 leverages a feed-forward 3DGS model for recovering the coarse geometry and appearance from sparse observations of a 3D scene, which are then used to render latent features as the pose and visual cues to guide the following SVD in generating 3D consistent 360° novel views. To demonstrate its effectiveness, we construct a challenging benchmark for 360° NVS of real-world scenes. Experimental results show that MVSplat360 achieves superior visual quality compared to other SoTA feed-forward approaches.

**Acknowledgements**    This research is supported by the Monash FIT Start-up Grant. Dr. Chuanxia Zheng is supported by EPSRC SYN3D EP/Z001811/1.

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

# A More Experiment Results

**Assessing the robustness of SVD's first-stage autoencoder to input resolution.** Since our MVS-plat360 fine-tunes the denoising process in latent space, it is essential to ensure the autoencoder accurately maps between pixel and latent spaces. We observe that when the input resolution ($256 \times 480$) differs significantly from the autoencoder's pre-trained resolution ($768 \times 1280$), the autoencoding process causes noticeable information loss, leading to missing details (see Fig. A 2nd column) with an average PSNR of 26.77dB on the test set. Although fine-tuning the autoencoder might address this limitation, it risks overfitting due to our relatively small-scale training set (around 2000 scenes). Instead, we find that simply upscaling the input $2\times$ via bilinear interpolation before feeding them to the encoder helps preserve details effectively (see Fig. A 3rd column) and improves the PSNR to 32.52dB, ensuring the latent space remains accurate.

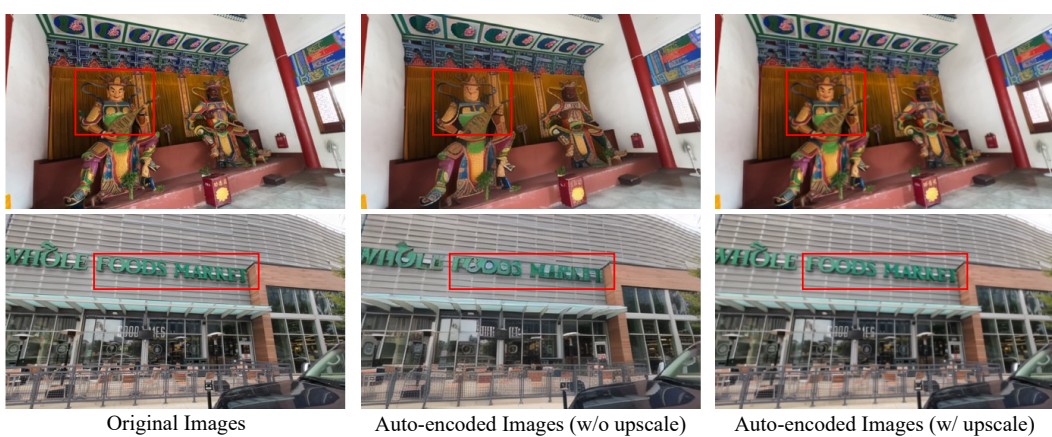

| Original Images | Auto-encoded Images (w/o upscale) | Auto-encoded Images (w/ upscale) |

Figure A: **Autoencoder using inputs with different resolutions**. The SVD's first-stage encoder and decoder are sensitive to image resolution. Hence, to ensure that images are encoded to the expected latent space, we upscale them 2 times via bilinear interpolation before feeding them to SVD.

# B More Implementation Details

Following MVSplat [10], we set the near and far depth range used in the cost volume construction as 1 and 100, respectively. We render features from the 3D Guassians rasterizer with shape $4 \times h \times w$, where $h$ and $w$ refer to the image height and width, respectively, and the channel dimension is set to 4 to align with that of the SVD initial conditional vector. We bilinearly interpolate the latent features to a resolution of $1/4h \times 1/4w$, matching the encoded features from images of size $2h \times 2w$, as the encoder downsamples inputs by a factor of 8. This design ensures proper projection into the initial latent space, as detailed in Appendix A. The SVD decoder outputs are then bilinearly interpolated from $2h \times 2w$ back to $h \times w$. Due to resource limitations, we mainly experiment on the "images_8" branch of the DL3DV-10K dataset, which contains images with resolution $256 \times 480$. Our default model is trained with the Adam optimizer, and the learning rate is set to $1.e - 5$ and decayed with the one-cycle strategy. All models are trained for 100,000 steps with an effective batch size of 8 on 1 to 8 A100 GPUs, and we apply the gradient accumulation technique whenever needed. During training, we sampled 5 views as input views and another 14 views as targeted rendering views, with the intention of better aligning with the following SVD module, which is trained on videos with 14 frames.

# C Limitations and Discussions

Although MVSplat360 achieves plausibly consistent 360° NVS and significantly outperforms previous works in visual quality by leveraging SVD, it also inherits several limitations from SVD. For instance, the results may exhibit oversaturated colors (see Fig. B, left), limiting improvements on pixel-aligned metrics like PSNR and SSIM. This is likely because SVD is primarily trained on artistic videos with vibrant colors, and fine-tuning from its pre-trained weight can bias the outputs toward the original

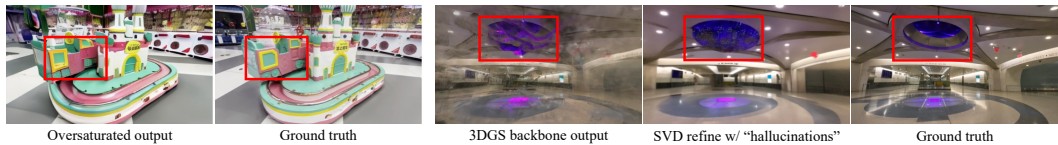

| | | | | | |
|---|---|---|---|---|---|
| Oversaturated output | Ground truth | | 3DGS backbone output | SVD refine w/ "hallucinations" | Ground truth |

Figure B: **Limitations**. Left: Views may appear oversaturated; Right: Refined views can contain over-hallucinated contents. Both limitations stem from the diffusion module's prior knowledge.

training data distribution [50, 30]. Additionally, the outputs might exhibit hallucinations and contain contents not existing in the input views (see Fig. B, right). Lastly, the inference is slow due to the multiple sampling steps in the diffusion process. We expect these limitations to be mitigated as better video diffusion models and pre-trained weights become available in the future.

## D    Broader Social Impacts

Our MVSplat360 renders 360° novel views from sparse observations, making it a valuable tool for augmented reality applications. It can enhance immersive experiences in entertainment and media, including 3D videos and video games, and support historical reconstruction for educational purposes. However, MVSplat360 's powerful generative capabilities could be misused to create fake videos. Additionally, while the rendered views are high quality, they may not fully capture real-world details. Therefore, precautions are necessary when using the generated data in safety-critical applications, such as training autonomous driving models.

## E    More Visual Comparisons.

Below, we provide more visual comparisons with existing state-of-the-art models.

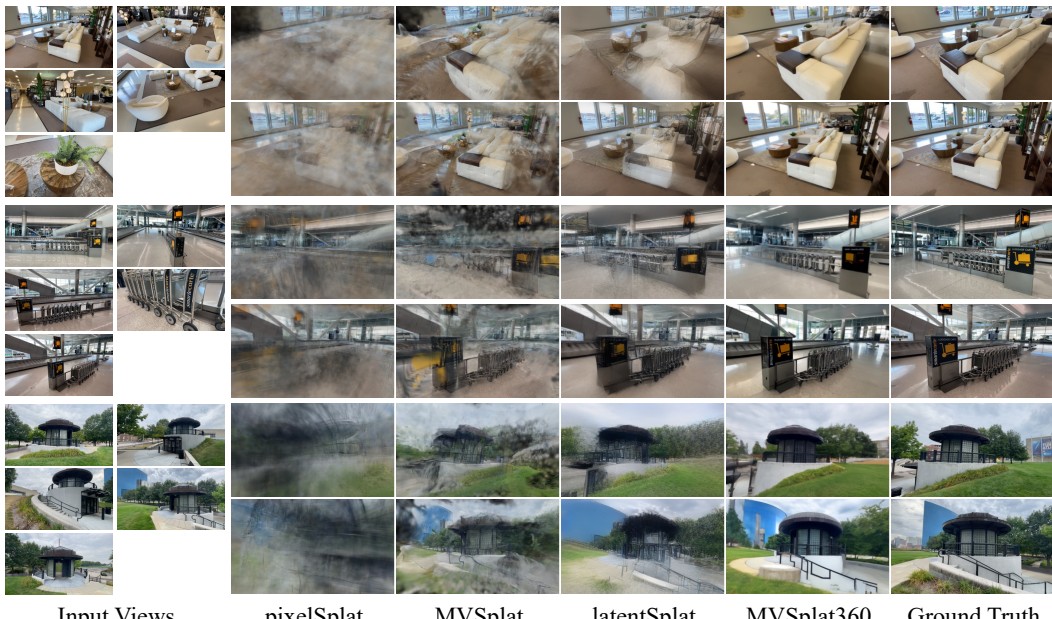

| Input Views | pixelSplat | MVSplat | latentSplat | MVSplat360 | Ground Truth |
|---|---|---|---|---|---|

Figure C: **More qualitative comparisons on DL3DV-10K**. Our MVSplat360 significantly outperforms all existing models in various complex scenes.

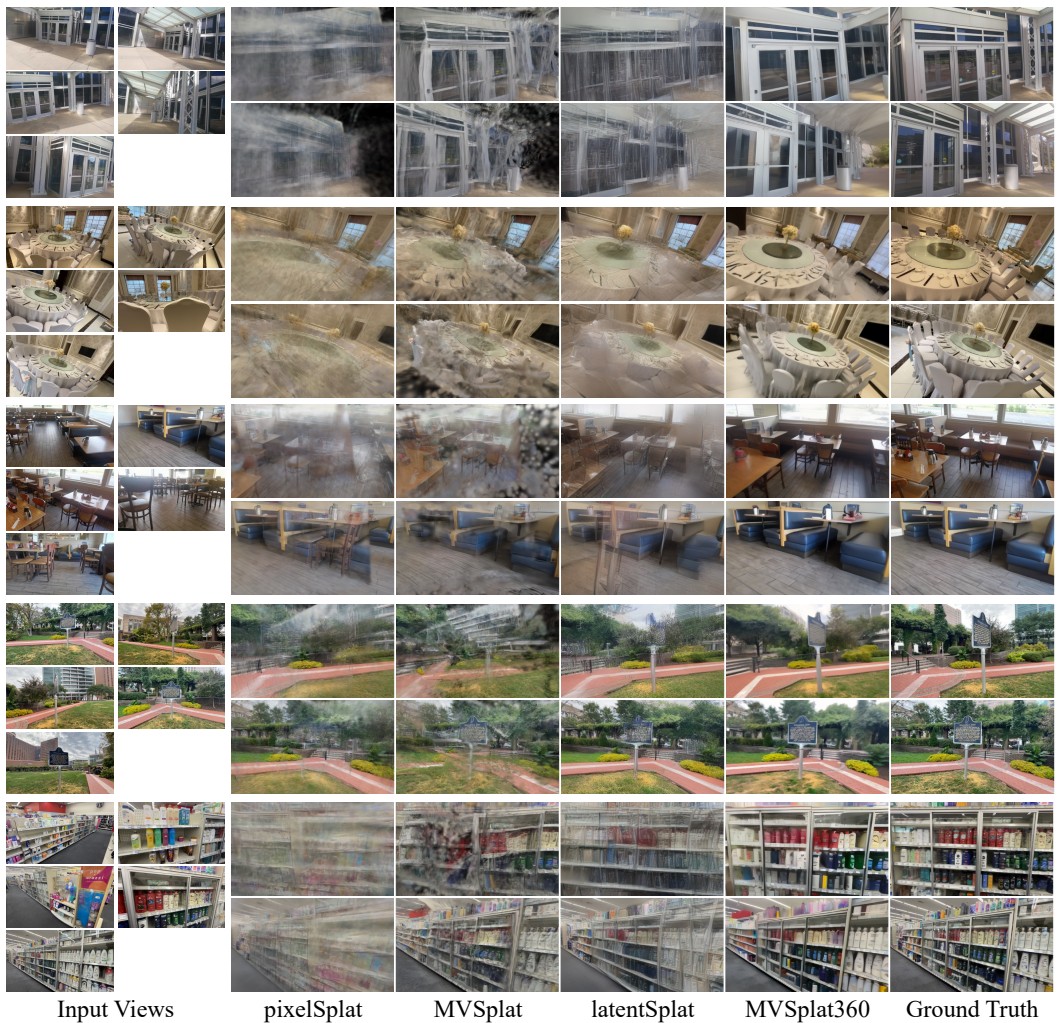

| Input Views | pixelSplat | MVSplat | latentSplat | MVSplat360 | Ground Truth |

Figure D: **More qualitative comparisons on DL3DV-10K**. Our MVSplat360 significantly outperforms all existing models in various complex scenes.

