# OpenReview forum: "MVSplat360: Feed-Forward 360 Scene Synthesis from Sparse Views"
_NeurIPS.cc/2024/Conference — NeurIPS 2024 poster_

### Official Review · Reviewer_3TXW · 2024-06-13

**Soundness:** 2
**Presentation:** 3
**Contribution:** 2
**Rating:** 5
**Confidence:** 4

**Summary:**

This paper proposes a framework of generalizable 3DGS for 360 degree of NVS. The framework comprises of two main components, one is similar to MVSplat that combines multi-view context image information, the other uses a pre-trained diffusion model for post-processing, which is conditioned on the rendered features from the MVS model and outputs detailed images. They also condition the diffusion model on the CLIP features of input images to integrate high-level semantics. Experiments show improvements on both DL3DV and RealEstate10K datasets, especially in qualitative results where the model can generate plausible contents that are not observed in the context images.

**Strengths:**

1. Originality: the paper proposes a novel setting of scene-level wide-sweeping representation through feed-forward. The setting is practical and with potential.

2. Clarity: The paper is well-written and readers can easily understand the proposed method.

**Weaknesses:**

1. Contribution: One limitation of this paper is the technical contribution. The combination of generative model and genalizable 3DGS, and conditioning it on rendered features, are similar to latentSplat [1], and the generative process is directly taken from video diffusion models [2]. And the multi-view image feature fusion is taken from MVSplat [3]. The key contribution is somewhat ambiguous.
2. Experiments: the reported quantitative improvements over previous SOTA methods are limited on both datasets. PSNR of $17.00\sim 18.21$ and SSIM of $0.496\sim 0.553$ is far from satisfactory. The authors mentioned that the refinement module does not guantee improvements on pixel-wise metrics, but the improvements on LPIPS is also minor.
3. Efficiency: another key limitation is on the rendering efficiency. Comparing to latentSplat, integrating a heavy diffusion model for post-processing leads to significantly slower rendering speed ($>100FPS\rightarrow\sim 1.75FPS$). The major problem is that applying similar post-processing as NeRF-based works like ReconFusion [4] largely hinders the real-time rendering of 3DGS.



[1] Wewer, Christopher, et al. "latentsplat: Autoencoding variational gaussians for fast generalizable 3d reconstruction." arXiv preprint arXiv:2403.16292 (2024).

[2] Blattmann, A., Dockhorn, T., Kulal, S., Mendelevitch, D., Kilian, M., Lorenz, D., Levi, Y., English, Z., Voleti, V., Letts, A., et al.: Stable video diffusion: Scaling latent video diffusion models to large datasets. arXiv preprint arXiv:2311.15127 (2023)

[3] Chen, Yuedong, et al. "MVSplat: Efficient 3D Gaussian Splatting from Sparse Multi-View Images." arXiv preprint arXiv:2403.14627 (2024).

**Questions:**

1. To compare with latentSplat more thoroughly, is it possible to evaluate the performance using MVSplat+latentSplat Decoder? The results could better evaluate the necessity of applying a heavy diffusion model for decoding.
2. PSNR results are missing in Table 3(b), and the performance is still not satisfying when using 7 views. Is it possible to provide ablation study results with PSNR using more input views ($>7$) during inference? The results can evaluate the generalization of the model to credibly reconstruct (nearly) fully observed scenes, and also strengthens the claimed 360$^{\circ}$ NVS that can possibly be applied to large scene reconstruction.
3. In Table 2, the extrapolation results of pixelSplat and latentSplat on re10k seem to be adopted from Table 2 of latentSplat, but latentSplat reported interpolation PSNR of 24.32dB for pixeSplat (instead of 25.89dB) due to a different training strategy (supervising on both interpolation and extrapolation views). Is it possible to report latentSplat results and pixelSplat extrapolation results on re10k using the same training strategy as the other methods? The results would form more fair comparisons.
4. In Table 2, MVSplat360 improves only 0.12dB PSNR on extrapolation results over MVSplat. However, there should be major cases where the non-generative methods render nothing in the unobserved regions in extrapolation views (eg. Figure 4 pixelSplat and MVSplat results) due to lack of post-processing. Theoretically, MVSplat360 should improves more than only 0.12dB over MVSplat.

**Limitations:**

See the weaknesses and questions above.

---

> ### Author Rebuttal · Authors · 2024-08-07
>
> ## **Response to Reviewer 3TXW (R4)**
>
> &nbsp;
>
> ### **Q1: Limited technical contributions.**
> A1: Kindly refer to the global response to all reviewers for more discussions regarding our contributions.
>
> &nbsp;
>
> ### **Q2: The scores of quantitative results on DL3DV are smaller than expected.**
> A2: These values are reasonable since the latest generative 3D synthesis is different to the traditional 3D reconstruction. The PSNR, SSIM, and LPIPS are all pixel-aligned metrics, which are not fully correct for the generated new content. Our MVSplat360 is able to generate multiple and diverse results for the occluded and invisible regions, which are consistent with high quality, but may not match the original ground truth. Note that our generated 360 video is consistent (supplementary video), and the results match well with the inputs’ distribution (FID score improves around 13.4)
>
> For better reference, we refer to the scores of Reconfusion (CVPR24) and CAT3D (arXiv24) tested on a similar dataset mip-NeRF 360. As reported in their paper, Reconfusion scores (PSNR=16.93, SSIM=0.401, LPIPS=0.544) and CAT3D scores (PSNR=17.72, SSIM=0.425, LPIPS=0.482). Note that even with per-scene optimisation, their scores are not impressive by using 6 observed views and averaging over only 9 scenes, while our scores are reported with a more challenging feed-forward manner, using only 5 observed views and averaging over 140 scenes.
>
> Nonetheless, our MVSplat360 outperforms all existing state-of-the-art models, as reported in Tab. 1. The extensive visual results in both the paper and the supplementary video also verify its superiority.
>
> &nbsp;
>
> ### **Q3: The rendering speed is slow.**
> A3: Unlike the latest related works, such as ZeroNVS, ReconFusion and CAT3D, that trained additional 3D NeRF for each scene, our MVSplat360 can directly generate consistent novel view video in a feed-forward architecture, which is much faster than these related models (see further discussions in the global response). As discussed in the limitation section (L517-L520), we are aware that integrating with a diffusion model will lead to slower rendering speed. This is caused by the multi-step denoising process, while it can be sped up by those improved one-step SVD models such as SF-V. Since this is beyond the main focus of our work, we leave it for future improvement. Besides, although latentSplat runs faster, its rendering quality on complex scenes is far from satisfactory, as demonstrated in the supplementary video, and latentSplat’s GAN-based decoder is unstable and can cause the model to collapse on complex scenes (L248-L249).
>
> * SF-V: Zhang, Zhixing, et al. "SF-V: Single Forward Video Generation Model." arXiv preprint arXiv:2406.04324 (2024).
>
> &nbsp;
>
> ### **Q4: Comparisons with “MVSplat Encoder + latentSplat Decoder”.**
> A4: We replace latentSplat’s encoder with the MVSplat’s encoder, and its performance is reported below.
>
> |  | FID↓ | DISTS↓ | LPIPS↓ | SSIM↑ | PSNR↑ |
> |---|:---:|:---:|:---:|:---:|:---:|
> | latentSplat | 37.68 | 0.234 | 0.439 | 0.469 | 16.68 |
> | MVSplat Encoder + latentSplat Decoder | 35.16 | 0.229 | 0.436 | 0.471 | 16.68 |
> | MVSplat360 | **20.17** | **0.172** | **0.425** | **0.496** | **17.00** |
>
> As reported, replacing latentSplat’s encoder with the MVSplat’s slightly improves the performance since its backbone features are enhanced. However, latentSplat’s GAN-based decoder still limits its performance in modelling complex scenes. Our MVSplat360 again outperforms this upgraded latentSplat across all metrics, assuring the superiority of our design.
>
> &nbsp;
>
> ### **Q5: Testing with more than 7 views.**
> A5: This work mainly targets the sparse view scenario, which is more challenging yet practical for casual users. Testing the feed-forward model (refer to the difference between feed-forward and per-scene tuning models in the global response) with a large number of views is not our primary focus, which is also quite challenging. The main reasons are: 1) Our backbone model MVSplat, similar to pixelSplat, predicts 3D Gaussians in a pixel-align manner, it becomes more difficult to correctly align Gaussians from different views as the view number increases, leading to more artifacts. The unreliable reconstruction results will then impact our refinement module, potentially leading to unexpectedly worse quality. 2) Our model is trained on 5 views, testing it with a significantly different number of views might lead to less effective performance. Hence, we only verify our model within a reasonable range (from 3 to 7 views) to ensure the correctness of our conclusions.
>
> &nbsp;
>
> ### **Q6: Inter- and extrapolation experiments on RE10K.**
> A6: We train two separate models for the RE10K experiments, one for interpolation and one for extrapolation. To maintain a fairer comparison, we retrain latentSplat’s model using the interpolation-based training strategy, and its performance is reported in the table below. Our MVSplat360 remains the best. We will update it to Tab. 2 in the updated version.
>
> |  | LPIPS↓ | SSIM↑ | PSNR↑ |
> |---|:---:|:---:|:---:|
> | pixelSplat | 0.142 | 0.858 | 25.89 |
> | latentSplat (retrained with interpolation strategy) | 0.139 | 0.851 | 25.53 |
> | MVSplat360 | **0.126** | **0.869** | **26.41** |
>
> &nbsp;
>
> ### **Q7: The PSNR improvement on RE10K is smaller than expected.**
> A7: As replied in A2, the generative 3D synthesis is different from the traditional “accurate” 3D reconstruction. MVSplat360 has a more obvious improvement in the feature-level metrics (L274-L277), although the gain is smaller in the pixel-aligned metric. This might be caused by 1) the oversaturated issues of the SVD model, as detailed in the limitation section (L509-L511); 2) the generative model only provides a reasonable guess for the unseen regions, which is not guaranteed for a pixel-align match with the original ground truth.

---

> ### Author Response · Authors · 2024-08-11
>
> Dear Reviewer 3TXW,
>
> Did we satisfactorily answer your questions? Would you like us to clarify anything further? Feel free to let us know, many thanks.
>
> Best regards,
> Authors of #2348

---

> > ### Comment · Reviewer_3TXW · 2024-08-13
> >
> > Thank you for your detailed rebuttal, which addressed my concerns on the experiments. I think overall this is a technically solid paper on an under-explored direction with good qualitative results. Since I still have concerns about the efficiency of leveraging a video diffusion model on the rendered Gaussian features, I will first raise my rating to 5-borderline accept, and discuss further with the other reviewers and AC in the later phase.

---

> ### Author Response · Authors · 2024-08-13
>
> We thank the reviewer for the follow-up comments. We are very grateful for your recognition of our key contribution: good visual quality in an under-explored setting of 360 scene synthesis.
>
> Below, we summarise information to justify the running efficiency, in case you, other reviewers and AC want to refer to it during the internal discussion.
>
> &nbsp;
>
> * **The task of feed-forward 360 scene synthesis from sparse views is very challenging.**
>   * Sparse views (e.g., 5 views in our experiments) provide only limited coverage of a scene, resulting in many unseen or occluded regions when rendering from 360-degree viewpoints. This causes significant challenges to previous feed-forward methods like pixelSplat, MVSplat and latentSplat. Prominent failures can be observed in all these methods, as demonstrated in our supplementary video.
>   * Another line of research, like ReconFusion and CAT3D, relies on additional per-scene optimization to achieve plausible results, which is an order of magnitude slower than our method (minutes vs. seconds (Ours)).
>   * In this paper, we demonstrate that high-quality feed-forward 360 scene synthesis can be achieved from sparse views, without any additional per-scene optimization.
>
> * **The efficiency of video diffusion models can be potentially improved with the latest techniques since video diffusion models are actively developed and advanced.**
>   * The main efficiency bottleneck of our method lies in the video diffusion model, which is slow since it requires the multi-step denoising process.
>   * However, we note that speeding up video diffusion models is an active topic and advanced techniques are emerging. For example, the recent SF-V model speeds up SVD with improved *one-step* denoising. Our model can benefit from such advancements.
>
> &nbsp;
>
> Overall, we provide a viable solution to the under-explored task of feed-forward 360 scene synthesis from sparse views, we believe our efficiency can be further improved with more advanced video diffusion models.
>
> Feel free to let us know if you would like us to clarify anything further. Many thanks.

---

### Official Review · Reviewer_pb1U · 2024-07-07

**Soundness:** 2
**Presentation:** 3
**Contribution:** 3
**Rating:** 5
**Confidence:** 4

**Summary:**

This paper aims to advance 360° novel view synthesis from sparse observations in wild scene scenarios. The key idea is to utilize the improved MVSplat for coarse geometry, refined by a stable video diffusion model to enhance appearance. This differs from prior work that, due to sparse viewpoint inputs, resulted in the ambiguous rendering of wide-sweeping or even 360° novel views. The proposed MVSplat360 method has been evaluated on challenging datasets, showing a marked enhancement over existing state-of-the-art techniques.

**Strengths:**

1. Originality and Significance
The concept of combining the generalizable GS with the SVD model is eminently logical; this innovative method effectively addresses the current limitations of feed-forward neural novel view synthesis techniques in 360° scene scenarios. By establishing a new benchmark for 360° scene reconstruction from sparse views, this paper proposes a novel direction for future research in the field.
2. Experimentation and Evaluation
The authors constructed a new benchmark using the challenging DL3DV dataset and performed extensive comparisons on the RealEstate10K dataset. The results consistently show that MVSplat360 outperforms existing state-of-the-art methods in both qualitative and quantitative assessments.
3. Presentation
The paper is overall well-written and the limitations have been sufficiently discussed in the supplementary.

**Weaknesses:**

1. From L193-201 and the provided demo, it becomes apparent that for complex scenes with occluded and invisible parts, influenced by the generative model, MVSplat360 may struggle to achieve stable multi-view consistency solely through "Multi-view conditions."
2. The article addresses the issue of oversaturated colors through a simple post-processing method; however, the effectiveness of this solution is not validated in the experimental and ablation sections.

**Questions:**

1. In L193-201, the "Multi-view conditions" exhibit limitations in maintaining multi-view consistency across scenes. Firstly, can CLIP embedding tokens of the original visible views guarantee a scene-level global description that guides an accurate denoising process? Additionally, for occluded and invisible parts (as depicted by the black pixels in the rendered results in Fig. 4), are the GS-rendered features filled with background values for these areas? How is multi-view consistency ensured in these regions? It is advisable for the authors to include visual results or analyses of rendered views for invisible parts from multiple perspectives of the same scene in the experiments or appendices.
2. The process of “Appearance Refinement” appears to be influenced solely by the Gaussian spherical harmonics parameters. In cases where the geometry inferred by MVS is inaccurate, what impact would this have on the denoising process of SVD?
3. The distinction between a) and b) in Fig.5 is unclear; it is advisable to revise the figure captions accordingly.

**Limitations:**

The authors have thoroughly outlined the limitations of their work as well as the potential negative societal impacts.

---

> ### Author Rebuttal · Authors · 2024-08-07
>
> ## **Response to Reviewer pb1U (R3)**
>
> &nbsp;
>
> ### **Q1: Unsatisfactory multi-view consistency in complex scenes.**
> A1:
> 1) The setting is extremely challenging: as verified by all provided visual results, MVSplat360 is the only approach that can provide reasonably good results on occluded and invisible regions compared to other state-of-the-art feed-forward models.
> Although concurrent work CAT3D might have a similar demo, it differs from our setting as it relies on per-scene optimization. Please refer to the global response for more discussions between feed-forward and per-scene tuning settings.
>
> 2) Long-sequence multi-view consistency for the diffusion model is still under exploration: built on top of the video diffusion model SVD, MVSplat360 maintains good multi-view consistency in typical sequence lengths such as 14 frames. It only becomes less satisfactory when the sequence gets significantly longer such as 56 frames in our testing. This shortcoming originates from our component SVD, and it can be addressed by replacing SVD with other released more powerful video diffusion models in the near future.
>
> 3) Our rendered novel views maintain reasonable multi-view consistency as verified by the SoTA reconstruction model: we run structure-from-motion on our rendered novel views, and the results confirm that they are multi-view consistent (see Fig. III in the one-page PDF).
>
> &nbsp;
>
> ### **Q2: Ablation of the post-processing operation.**
> A2: Below is the comparison on DL3DV regarding with and without (w/o) post-processing operation, using the same setting as Tab.3.
>
> |                                 |  FID$\downarrow$  | DISTS$\downarrow$ | LPIPS$\downarrow$ |  SSIM$\uparrow$ |  PSNR$\uparrow$ |
> |---------------------------------|:-----:|:-----:|:-----:|:-----:|:-----:|
> | MVSplat360  w/o post-processing | 20.25 | 0.174 | 0.427 | 0.474 | 16.70 |
> | MVSplat360                      | 20.17 | 0.172 | 0.425 | 0.496 | 17.00 |
>
> As reported, the post-processing (histogram matching) helps align the output color space with the input one, hence relieving the oversaturated issues caused by SVD and improving the quantitative performance. The improvements are more obvious in those image space metrics (PSNR and SSIM) since other metrics are measured in the feature domain and are more robust to image space distortion.
>
> &nbsp;
>
> ### **Q3: More details of multi-view conditions**
> A3:
> 1) The CLIP embedding extracted from the context views (observed viewpoints) mainly aims to provide a high-level semantic understanding of the scene (L193-L195). But more importantly, the diffusion module is also conditioned with the 3DGS rendered features of the novel viewpoints (L197-L199). These features contain pose and texture information, guaranteeing a successful denoising process.
>
> 2) For invisible parts, the 3DGS rendered features are filled with background values since GS is a regression model with no generative power. Based on the visible parts, information in these invisible regions can be hallucinated by the diffusion model, which has strong prior knowledge gained from large-scale data. The multi-view consistency in the in-painted/out-painted areas is ensured by the strong prior of the video diffusion model. Related novel views containing invisible regions are provided in Fig. II of the one-page PDF.
>
> &nbsp;
>
> ### **Q4: The impact of the reconstructed geometry on SVD.**
> A4: The SVD refinement module is influenced by the rendered features, whose rendering/rasterization process takes as input all Gaussian attributes, including both geometry and texture factors, not solely the SH coefficients. In case the geometry is not accurate, the SVD model can help refine to get better visual results using its powerful prior knowledge.
>
> &nbsp;
>
> ### **Q5: Unclear illustration of Fig. 5**
> A5: In the updated version, we will indicate (a) and (b) in Fig. 5.

---

> ### Author Response · Authors · 2024-08-11
>
> Dear Reviewer pb1U,
>
> Did we satisfactorily answer your questions? Would you like us to clarify anything further? Feel free to let us know, many thanks.
>
> Best regards,
> Authors of #2348

---

### Official Review · Reviewer_RhQM · 2024-07-13

**Soundness:** 3
**Presentation:** 3
**Contribution:** 2
**Rating:** 5
**Confidence:** 3

**Summary:**

This paper proposes MVSplat360, a generalized sparse-view novel view synthesis method. MVSplat360 utilizes the Stable Video Diffusion model to guess the novel views besides input sparse views.

**Strengths:**

1. MVSplat360 achieves better novel view synthesis with sparse input views by introducing the stable video diffusion.
2. The paper is well-written.

**Weaknesses:**

1. Most of the success is due to the stable video diffusion, and this paper provides mainly engineering works.

**Questions:**

1. I would like to see what the contributions are besides introducing the diffusion model.

**Limitations:**

The authors have discussed the limitations.

---

> ### Author Rebuttal · Authors · 2024-08-07
>
> ## **Response to Reviewer RhQM (R2)**
>
> &nbsp;
>
> ### **Q1: Limited contributions: engineering via adding SVD**
> A1: Please refer to the global response to all reviewers for more detailed discussions of our main contributions.

---

> ### Author Response · Authors · 2024-08-11
>
> Dear Reviewer RhQM,
>
> Did we satisfactorily answer your questions? Would you like us to clarify anything further? Feel free to let us know, many thanks.
>
> Best regards,
> Authors of #2348

---

### Official Review · Reviewer_4RiL · 2024-07-26

**Soundness:** 3
**Presentation:** 2
**Contribution:** 2
**Rating:** 5
**Confidence:** 5

**Summary:**

The paper proposes MVSplat360, a method for wide-sweepign or 360-degree novel view synthesis on general scenes from sparse input views. It extends an existing state-of-the-art approach MVSplat to render 3D feature Gaussians as conditioning for a refinement network in form of a pre-trained video diffusion model, which is fine-tuned jointly. The approach is evaluated on RealEstate10k, an existing benchmark of home walkthrough videos, and on a newly proposed benchmark for 360-degree novel view synthesis, leveraging an existing dataset of diverse scenes. Quantitative and qualitative results show improvements over regression-based and generative, splatting-based baselines and plausable completions in case of incomplete observations.

**Strengths:**

- The paper tackles a challenging and interesting task of generalizable 360-degree novel view synthesis from sparse views on diverse real-world scenes.
- The proposed method combines the strengths of splatting-based generalizable 3D reconstruction and large-scale pre-trained video diffusion models as a genrative refinement network:
  - pixelSplat [1] and MVSplat [2] have shown strong performance in view interpolation.
  - latentSplat [3] has shown the advantages of using a decoder generative network.
  - Video diffusion models like Stable Video Diffusion [4] have been trained on massive data and learn a strong prior useful for plausible view extrapolation and scene completion.
- The problem definition and main approach explained well.
- The evaluation validates the effectiveness of MVSplat360
  - The proposed benchmark on the DL3DV dataset [5] is challenging and supports the claims of wide-sweeping or 360-degree novel view synthesis on large real-world scenes.
  - The approach outperforms all baselines in almost all metrics quantitatively on DL3DV and RealEstate10k.
  - Qualitative results show high-quality novel views with less artifacts than baselines for both datasets with plausible completions for extrapolation on RealEstate10k.
  - The supplement provides more convincing qualitative results including a video.
- The paper includes an ablation study validating all the proposed design choices and showing desired performance scaling w.r.t. increasing numbers of input views.

1. pixelSplat: 3D Gaussian Splats from Image Pairs for Scalable Generalizable 3D Reconstruction, CVPR 2024
2. MVSplat: Efficient 3D Gaussian Splatting from Sparse Multi-View Images, ECCV 2024
3. latentSplat: Autoencoding Variational Gaussians for Fast Generalizable 3D Reconstruction, ECCV 2024
4. Stable Video Diffusion: Scaling Latent Video Diffusion Models to Large Datasets, arxiv 2023
5. DL3DV-10K: A Large-Scale Scene Dataset for Deep Learning-based 3D Vision, CVPR 2024

**Weaknesses:**

- Confusing title: Given the contributions, the title should rather focus on the proposed method.
  - The title of the paper suggests a focus on benchmarking existing methods.
  - The paper still proposes a new approach MVSplat360.
  - Evaluation is done on one existing benchmark (RealEstate10k) and one newly proposed one leveraging the already existing DL3DV dataset.
  - The problem of generalizable novel view synthesis given sparse views is not novel (see baselines [1], [2], [3]) such that the benchmark creation only consists of the definition of input and target views.

- Insufficient contextualization relative to prior work:
  - Inaccurate use of the term 'concurrent work':
    - In line 60, ReconFusion [4] and latentSplat [3] (again in line 244) are referred to as concurrent work.
    - This is not really the case anymore, because
      - ReconFusion [4] appeared on arxiv in the beginning of December 2023 and was presented at CVPR 2024.
      - latentSplat [3] appeared on arxiv end of March 2024, 3 days after MVSplat [2], which is one of the building blocks of the proposed method.
  - Missing references to related work regarding conditioning via feature rendering and CLIP embeddings:
    - In lines 163-168 and 196-201, the authors propose to render latent features as spatial conditions for a generative model (Stable Video Diffusion), which is later ablated in lines 296-299.
       - This idea is not novel and the paper is missing important references in this context:
         - GeNVS [5] proposed to use rendered pixelNeRF features as conditoning for a 2D diffusion model.
         - ReconFusion [4] adopts this to condition a text-to-image latent diffusion model.
         - latentSplat[3] also renders feature maps that are decoded to a novel view in a GAN setup.
    - In lines 193, the authors describe the use of CLIP image embeddings of the input views as global conditioning.
      - ReconFusion [4] desribes a very similar procedure.

- Lack of clarity regarding the proposed method:
  - From the paper and supplement, it is not clear, for which loss terms the architecture is optimized during training.
    - Since the paragraph about "Multi-frame diffusion model" starting in line 174 mainly describes SVD [6], it is not clear, whether the loss in equation (2) is also the only loss for MVSplat360.
  - In line 167f., the authors claim that joint training of MVSplat and SVD can further enhance geometry through the feature conditioning.
    - Is that the only source of gradient signals to the structural parameters of the 3D Gaussians (location, scale, rotation...)?
    - ReconFusion [4] and latentSplat [3] both additionally optimize direct RGB renderings to regress the ground truth for better geometry gradients.
  - The role of the video diffusion model compared to a single-view diffusion model is not clear.
    - The paper misses to explain, whether and if so how novel views are refined jointly (see questions).

- The proposed approach is incremental.
  - Compared to ReconFusion, MVSplat360 uses MVSplat instead of pixelNeRF and a video instead of an image diffusion model.
  - Compared to latentSplat, it replaces the pixelSplat encoder with the improved MVSplat and a small CNN-based decoder with SVD [6].

- Unclear conclusion of benchmarking:
  - latentSplat builds upon pixelSplat, which is shown to be worse in depth estimation than MVSplat.
    - Regarding this baseline, it is difficult to conclude, how much the reconstruction and the refinement modules contribute to the performance.
  - If the goal of the paper is benchmarking, a component-wise evaluation (backbone feature extractors, depth estimation, refinement networks)  would be more insightful.

1. pixelSplat: 3D Gaussian Splats from Image Pairs for Scalable Generalizable 3D Reconstruction, CVPR 2024
2. MVSplat: Efficient 3D Gaussian Splatting from Sparse Multi-View Images, ECCV 2024
3. latentSplat: Autoencoding Variational Gaussians for Fast Generalizable 3D Reconstruction, ECCV 2024
4. ReconFusion: 3D Reconstruction with Diffusion Priors, CVPR 2024
5. GeNVS: Generative Novel View Synthesis with 3D-Aware Diffusion Models, ICCV 2023
6. Stable Video Diffusion: Scaling Latent Video Diffusion Models to Large Datasets, arxiv 2023

**Questions:**

- Are the structural parameters of the Gaussians purely trained via the SVD loss through the features?
  - Does the for features extended 3D Gaussians rasterizer compute gradients through the features to the structural parameters of the Gaussians?
  - If so, what is the intuition why this apparently works, while baselines (latentSplat and ReconFusion) use auxiliary losses?
- Are novel views refined jointly by the video diffusion model?
  - If yes, how do you deal with long videos?
  - Is temporal ordering leveraged or is the model agnostic to temporal permutations of target camera poses?
  - What are implications regarding 3D consistency?
- How is the training done on RealEstate10k?
  - Do you train two different models (also for baselines) for inter- and extrapolation?

**Limitations:**

The authors addressed limitations and societal impacts in the appendix.

---

> ### Author Rebuttal · Authors · 2024-08-07
>
> ## **Response to Reviewer 4RiL (R1)**
>
> &nbsp;
>
> ### **Q1: “Benchmarking” in the paper title is inaccurate.**
> A1: To make the title reflect our contributions more accurately, we decided to change the title to “MVSplat360: Feed-Forward 360 Scene Synthesis from Sparse Views”.
> Our initial intention is to emphasise that MVSplat360 is the first to explore a new setting: 360-degree feed-forward NVS from sparse views for large-scale scenes, and we demonstrate how to benchmark the models’ effectiveness in this setting. Although it differs from the existing two dominated settings (360-degree NVS for objects or limited NVS for scenes) only in source and target view selection, none of the existing approaches can achieve plausible outcomes in this new setting in a feed-forward way (Fig.3, Fig. 6 and the supplementary video).
>
> &nbsp;
>
> ### **Q2: Discussions of related “concurrent” work**
> A2: We acknowledge that MVSplat360 is built on top of existing components, similar to several related works (L120, L191-192), but the key focus of this work is to verify that MVSplat360 renders remarkably higher-quality images in an unexplored new setting: 360-degree feed-forward NVS from sparse views for large-scale scenes.
> Existing approaches, such as ZeroNVS, Reconfusion and CAT3D, still need to train a NeRF with the per-scene optimization approach (L61). GeNVS is tailored for object-centric scenes. LatentSplat fails to render plausible views in the new setting (Fig. 3 and Fig. 6), and its GAN-based decoder easily leads to model collapsing (L248).  We will add detailed discussions in the updated version and remove the term “concurrent work” to avoid unintentional misunderstandings.
>
> &nbsp;
>
> ### **Q3: More details of MVSplat360**
> A3:
> * **Training objectives:** Similar to latentSplat, our MVSplat360 contains two objective functions: reconstruction loss and diffusion loss. We modify the MVSplat backbone to render 1) images and 2) features. 1) The images are supervised by the reconstruction loss (L1 and LPIPS loss) against the ground truth RGB images. 2) The features are used as conditions in the following SVD module supervised by the diffusion loss (MSE in latent space as detailed in L185-188). We will detail the training objective functions in the updated version, and will also release our code to ensure better understanding.
>
> * **SD v.s. SVD**: In our experiments, we observe that refining using the single-view diffusion model leads to inconsistency among different novel views, which is similar to the findings of Reconfusion (see Fig. 4 in Reconfusion’s paper). On the other hand, we do not want to train additional 3D NeRF/GS for each scene (like Reconfusion did), which is expensive and fussy. To achieve consistent novel view synthesis without additional per-scene training, we therefore opt for the video diffusion model. The integration is achieved by channel-wise concatenating a sequence of the rendered features with the sampled noise (L196-L198).
>
> &nbsp;
>
> ### **Q4: The approach is incremental.**
> A4: Kindly refer to the global response to all reviewers for more discussions regarding our contributions.
>
> &nbsp;
>
> ### **Q5: Conclusion of “benchmarking”: ablations on reconstruction and refinement modules**
> A5: Regarding the unintentional misuse of the term “benchmarking,” kindly refer to A1.
> 1. We provide comparisons with the “MVSplat Encoder + latentSplat GAN Decoder” as suggested by R4Q4, which further confirms the superiority of our design.
> 2. We refer the readers to MVSplat’s paper for the ablations of backbone feature extractors and depth estimation. The ablations of the refinement module are provided in Tab. 3.
>
> &nbsp;
>
> ### **Q6: Training objectives of the 3DGS reconstruction module**
> A6: Kindly refer to "Training objectives" in A3.
>
> &nbsp;
>
> ### **Q7: More details of the refinement module SVD.**
> A7:
> 1. The SVD module can, by default, generate an arbitrary number of views. In our experiments, during training, we fixed the target rendering view number to 14 to better align with the SVD pre-trained model (L502-L503). At inference, we feed 56 views to the SVD module and change all related temporal attention blocks to local attention with a window size of 14 to better align with our training.
> 2. Our trained refinement module is agnostic to temporal permutations. The main reason is that we condition the diffusion model with features rendered by the 3DGS from arbitrary viewpoints, while related work such as SV3D and CAT3D conditions on camera trajectory, leading to their requirements of meticulously designed order (L135-L140).
> 3. 3D consistency implies that a) our model is 3D-aware due to the usage of 3DGS; b) the rendered novel views are multi-view consistent thanks to the video diffusion module. In addition, we run structure-from-motion on our refined novel views, whose results further confirmed that our outputs are 3D consistent (see Fig. III in the one-page PDF).
>
> &nbsp;
>
> ### **Q8: Inter- and extrapolation experiments on RE10K.**
>
> A8: Kindly refer to R4Q6.

---

> ### Author Response · Authors · 2024-08-11
>
> Dear Reviewer 4RiL,
>
> Did we satisfactorily answer your questions? Would you like us to clarify anything further? Feel free to let us know, many thanks.
>
> Best regards,
> Authors of #2348

---

> > ### Comment · Reviewer_4RiL · 2024-08-11
> > **Rebuttal Questions**
> >
> > I thank the authors for their clarifications regarding my concerns and providing additional results. Follwing the rebuttal, I still have some questions and concerns:
> >
> > > **A2: Discussions of related “concurrent” work**
> >
> > The discussion regarding related work is not completely accurate. GeNVS is not limited to object-centric scenes, but the paper includes results for the Matterport3D dataset, which is very similar to RealEstate10k. Moreover, ZeroNVS, ReconFusion and CAT3D do not necessarily require per-scene optimization of a NeRF, but can also be used directly for possibly inconsistent novel view synthesis. However, this is exactly comparable to the proposed MVSplat360. For obtaining a consistent 3D representation that allows fast rendering, the output novel views would need to be fused via optimization, e.g., of 3D Gaussians or a NeRF.
> >
> > > **A3: Training objectives**
> >
> > Could you elaborate more on my first question (and sub-points) in the questions section of my review?
> > In line 167f. of the paper, you claim that joint training of MVSplat and SVD can further enhance geometry through the feature conditioning. Is that really the case, if the auxiliary reconstruction loss is the only source of gradients for the structural parameters of the Gaussians?
> >
> > > **A3: SV vs SVD**
> >
> > To me, the fact that you denoise novel views jointly was not clear from the paper. The comparison with SD sounds like an interesting insight that would be very nice to have in the paper as an ablation. However, I have some follow-up questions regarding this:
> > - As you use concatenation along the channel dimension processed by a 3D UNet of SVD, I would expect problems if novel views are too far away from each other w.r.t. the camera pose. Is that the case?
> > - How do you handle this at test time?
> >   - Does a set of novel views always have to be a plausible camera trajectory?
> >   - How long are these trajectories?
> > - Would cross-view attention as done in multi-view diffusion models be a more suitable alternative than concatenation in channel dimension?
> >
> > > **A7: More details of the refinement module SVD**
> >
> > 1. SVD does not only consist of temporal attention, but also temporal convolutions. Is temporal consistency in the camera trajectory considered for this fact?
> > 2. I am confused about this. If I am not mistaken, CAT3D conditions on individual camera poses, but does not assume any consistent temporal camera trajectory for this. SVD however is build for videos, e.g., by using temporal convolutions. Therefore, the architecture is tailored for temporally consistent camera poses, making it not agnostic to temporal permutations. Please correct me, if I am wrong.

---

> ### Author Response · Authors · 2024-08-12
> **Further response to follow-up comments (1/2)**
>
> We are grateful to see the follow-up comments. We address the additional concerns below. Feel free to let us know if you would like us to clarify anything further.
>
> > **A2: Discussions of related “concurrent” work**
>
> **Description of GeNVS**
>
> We agree that the Matterport3D dataset used by GeNVS is similar to RealEstate10K, and we will correct its description to “GeNVS mainly works on 360-degree object-centric scenes *or nearby viewpoint scene-level datasets*”. Nonetheless, it does not alter the previous conclusion as it still belongs to the two existing scenarios summarized in L28-L30 and Fig. I of the one-page PDF. The key contribution of MVSplat360 is that it focuses on an *unexplored* setting: feed-forward *360-degree* scene synthesis. The *majority of experiments are conducted on DL3DV* (rather than RealEstate10K), which significantly verifies MVSplat360’s effectiveness in handling 360-degree NVS from sparse views.
>
> **Discussions with ReconFusion**
>
> * We refer the reviewer to the project page of ReconFusion for the results from the diffusion model. In particular, as shown in the last video entitled “ReconFusion distils a consistent 3D model from inconsistent samples” on ReconFusion’s project page, views sampled purely from its diffusion module are far from consistent, showing obvious jittering from frame to frame. Our supplementary video shows that MVSplat360 renders multi-view novel views with much higher consistency.
> * Note that those inconsistent demos provided by ReconFusion mainly consist of forward-facing or *constraint* orbital camera trajectories, while our consistent demo contains way more challenging rendering trajectories, including different types of *unconstrained* trajectories.
> * It is necessary for ReconFusion to apply per-scene optimization in order to get satisfactory consistent novel views, while our MVSplat360 renders 3D consistent novel views in a feed-forward manner thanks to our effective design summarised in “Contributions of MVSplat360 (Method)” in the global response to all reviewers.
>
> &nbsp;
>
> > **A3: Training objectives**
>
> We will update this potentially ambiguous claim regarding "enhancing geometry" in the paper.
>
> * It is correct that the reconstruction loss is the only source of gradients for the structural parameters of the Gaussians.
> * Our initial intention is that joint training can help “enhance the backbone features”. Since the features and other Gaussian parameters are from different heads but share the same backbone, we assume that enhancing the backbone feature will lead to a better reconstruction module and, hence better reconstructed coarse geometry. To avoid unintentional overclaim, we will tone down L168 to “the SVD loss can further optimize the Gaussian features, further *enhancing the reconstruction backbone*”.
>
> &nbsp;
>
> > **A4: SD vs SVD**
>
> **Ablation of SD**
>
> In the updated version, we will make it clearer that we denoise all novel views jointly. We will also add visual comparisons with the SD-based ablation model, which shows inconsistent novel views compared to our default design, as observed in our experiments.
>
> **When novel views are far away from each other**
>
> We did not observe obvious limitations regarding this issue. This is probably because although the viewpoints range from 180 to 360 degrees for each scene in DL3DV, it still belongs to one scene and might not contain significant “far away” novel views.
>
> **Test time trajectories**
>
> * In our experiments, the set of novel views is always a plausible trajectory. In particular, for quantitative evaluation, we use the camera trajectories captured by the initial video. For the video demo, we apply a Gaussian filter to the initial captured camera path to obtain 6DoF stabilization results.
> * Each trajectory contains 56 frames, which are uniformly sampled from the initial video that contains around 300 frames.
>
> **Cross-view attention vs. concatenation**
>
> In typical single-view or multi-view diffusion-based NVS models, their input features come from the *source/observed viewpoints*. In this case, they are required to align/correct those source viewpoint features to match with the novel view cameras, hence it might be better to achieve via cross-view attention. In contrast, in MVSplat360, the features provided to the SVD are those rendered from *target/novel viewpoints*, containing coarse but correct geometry information for the target viewpoint, hence it is reasonable to achieve via concatenation.
>
> It might be helpful to add *additional source/observed view* features to the SVD refinement module via cross-view attention, keeping the current concatenation for *target/novel views*. We will explore this strategy in our further experiments. Thanks for the insightful suggestion!
>
> &nbsp;
> &nbsp;
>
> *(see next comments for more response, thanks.)*

---

> ### Author Response · Authors · 2024-08-12
> **Further response to follow-up comments (2/2)**
>
> *(see previous comments for more response, thanks.)*
>
> &nbsp;
> &nbsp;
>
> > **A7: More details of the refinement module SVD**
>
> **Temporal convolution**
>
> We do not apply similar operations to temporal convolution. We observe that applying temporal attention to all 56 frames leads to oversmoothness, so we change it to local attention with a window size of 14. In contrast, convolution is operated in local regions and has no such issues.
>
> **Discussions with CAT3D**
>
> Sorry for the unintentional confusion regarding “permutation.” Our initial intention was to emphasize that our refinement module takes *unconstrained* camera trajectories. We assume that the trajectory is plausible, as detailed in the above clarification regarding A4 (Test time trajectories), but we do not require it to be a meticulous design such as an orbital trajectory.
>
> In contrast, CAT3D does assume that its camera trajectory should meet several strict requirements, as detailed in its paper Sec. 3.2 and Appendix C.
>
> &nbsp;
>
> *Feel free to let us know if you would like us to clarify anything further. Many thanks.*

---

> > ### Comment · Reviewer_4RiL · 2024-08-13
> >
> > Thanks again for the discussion. The rebuttal addresses most of my concerns such that I would like to increase my rating to 5: borderline accept.
> > Reasons for that are:
> > - I agree with the significance and difficulty of the 360° NVS setting from sparse views on scene level and the strong performance of the proposed method on the DL3DV benchmark.
> > - The discussion resolved the lack of clarity in the paper.
> >
> > Additionally, I would like to give the following suggestions for a final version:
> > - The joint generation of novel views using a video (SVD) instead of an image diffusion model (e.g. SD) conditioned on features rendered from a 3D representation and the resulting 3D consistency is a very important part of the method and should be highlighted and further evaluated.
> >   - Regarding multi-view diffusion models as a recent trend, 3D consistency of the generated views seems to be the main bottleneck.
> >   - It would be very interesting to see how view-conditioning via features rendered from a 3D representation compete against alternatives like Plücker coordinates used in multi-view diffusion models.
> > - For a more convincing evaluation of 3D consistency (compared to the SfM results in the rebuttal PDF), I would recommend a similar approach of mesh reconstruction from generated novel views as done in latentSplat.
> >
> > The incremental nature of the proposed method compared to previous works is the main reason that prevents me from giving an even higher rating.

---

> > > ### Author Response · Authors · 2024-08-13
> > >
> > > We thank the reviewer for the thoughtful discussions. We are more than grateful for your recognition of our key contributions: our strong performance on the under-explored and challenging 360 scene synthesis setting.
> > >
> > > We will make the writing clearer following all of our discussions. We will also start working on your follow-up suggestions regarding highlighting and further evaluating SVD in terms of 3D consistency, the comparison with Plücker coordinates conditions and mesh reconstruction. Thanks again for your insightful suggestions for making this work more solid.

---

### Author Rebuttal · Authors · 2024-08-07

## **Global Response to All Reviewers**

&nbsp;

We thank all reviewers for their constructive comments. We are encouraged by the appraising comments "the problem definition and main approach explained well", "the evaluation validate the effectiveness of MVSplat360" (**4RiL**), "achieves better NVS with sparse input views" (**RhQM**), "a novel direction", "better performance" and "well written" (**pb1U**, **3TXW**).

&nbsp;

Please kindly check the **attached one-page PDF** for more information. In particular, the PDF contains

* **Figure I: Taxonomy of existing sparse view novel view synthesis.** Our MVSplat360 is the first to address an unexplored task: feed-forward 360 scene synthesis from sparse views.
* **Figure II: Visualization of our MVSplat360 generates multi-view consistent content for invisible regions.**
* **Figure III: 3D reconstruction using input and rendered views from our MVSplat360.** We obtain reasonably good 3D reconstructions of a 360-degree scene, indicating that the rendered views from our model are multi-view consistent and geometrically correct.

&nbsp;

We provide more detailed discussions below.

### **1: Taxonomy of sparse view novel view synthesis**
A1: Although sparse view novel view synthesis has been extensively explored in recent years, our MVSplat360 targets an unexplored new task. Below, we present the taxonomy regarding our related work (readers are recommended to see Fig. I in the one-page PDF for a more expressive diagram version).

```
|---- Sparse view novel view synthesis
      |---- Per-scene optimization (Reconfusion, CAT3D, ZeroNVS, etc.)
      |---- Feed-forward / Generalizable
            |---- Object-centric
                  |---- 360-degree viewpoints (Zero123, Free3D, SV3D, LGM, latentSplat, etc.)
            |---- Scene-level
                  |---- Nearby viewpoints (pixelSplat, MVSplat, latentSplat, etc.)
                  |---- 360-degree viewpoints (MVSplat360)
```

Note that both Reconfusion and CAT3D require per-scene optimization in a two-step pipeline, where they first generate dense views from a multi-view diffusion model and then optimize a NeRF for each specific scene. In contrast, our MVSplat360 directly outputs 360-degree views in a single feed-forward inference without any test-time optimization. Since Reconfusion and CAT3D require optimizing a NeRF for every unseen scene, it leads to significant time (10+ mins/scene) and additional storage (100+ M/scene) consumption. In contrast, our MVSplat360 uses only a single model, and it can be directly tested on unseen scenes, which is more efficient in terms of inference time (~32 secs/scene) and storage consumption (0 additional storage/scene).

&nbsp;

### **2: Contributions of MVSplat360**

A2: Although existing works, such as GeNVS, ZeroNVS, Reconfusion, and CAT3D, also train/fine-tune a diffusion generator for novel view synthesis, the research goal and technical difference are significant:

1) **(Task) an unexplored yet more practical setting:** 360-degree feed-forward NVS from sparse views for large-scale scenes is an unexplored setting. Most highly related works, such as ZeroNVS, Reconfusion and CAT3D, need to train a NeRF for each scene (which are *not* feed-forward), while other 360-degree NVS works mainly focus on object-centric scenes (L27-L30).
In contrast, our main goal is to build a feed-forward 360-degree NVS model for large-scale scenes without per-scene optimization. To the best of our knowledge, we are the first to explore this challenging new setting on the new dataset, which will shed new light on how to advance the area of sparse-view NVS.
2) **(Method) an effective feed-forward model for the new setting:** Although combining conditional NeRF with an image/video generator has been explored in recent works, coming up with our design and making it work for such a challenging setting are non-trivial. All existing SoTA approaches fail to achieve satisfying results on the new setting. In particular, we 1) build a GS feature rendering as coarse geometry for multiview SVD generation; 2) extend the single-view-to-video SVD model to a multi-view conditioned refinement model; 3) propose a color adjustment mechanism to relieve the oversaturated issue in SVD.
Furthermore, to achieve the goal, it needs suitable training data and training strategy. We build a new training and testing split from the latest DL3DV dataset and design the nearest view warping strategy to train the model.
Note that many SoTA impactful methods are also simple at the concept level, e.g., Splatter Image and pixelSplat replace the NeRF in pixelNeRF with 3DGS, and MVSplat replaces the NeRF in MVSNeRF with 3DGS, but similar to ours, a lot of detailed designs are needed to make them work.
3) **(Results) remarkably higher quality visual results.** As demonstrated in the paper and the supplementary video, MVSplat360 renders much better novel views than all existing SoTA feed-forward models. We highly recommend the readers to view the supplementary video.

---

### Decision · Program_Chairs · 2024-09-25

**Decision:**

Accept (poster)

**Comment:**

The paper received borderline reviews initially, stating a large variety of concerns about the clarity, paper framing and technical novelty of the approach. At the same time, all reviewers acknowledged the high quality results of the presented method in a very challenging setup.

The authors provided exhaustive answers, leading to the scores converging to 4x borderline accept. While some concerns regarding clarity and paper framing have been resolved, some others, especially technical novelty, remain unresolved.

After reading the paper, reviews and rebuttal, I can draw the following conclusions:
- The paper indeed shows impressive results in 360° novel view synthesis, qualitatively and quantitatively.
- The tackled setting is a very challenging one and still heavily underexplored. Methods that solve it are very welcome in the community.
- The technical novelty is limited, as the method essentially combines two existing methods (MVSplat + SVD) into a single pipeline in a similar way as Reconfusion did with pixelNeRF + an image diffusion model.
- The paper could do a better job in highlighting specific aspects that surfaced during the discussion phase, such as the effect of using a pre-trained video diffusion model instead of image diffusion models. Such analysis could lead to more conceptual insight transported to the reader, strengthening the paper.

This paper is clearly in a borderline state. Since all reviewers are agreeing on an accept score and I am also impressed by the qualitative results of the work, I follow the reviewers and recommend acceptance.

I heavily encourage the authors to adjust the paper by incorporating the different aspects that surfaced in the discussion phase.